# NEW PERSPECTIVE ON THE GLOBAL CONVERGENCE OF FINITE-SUM OPTIMIZATION

## ABSTRACT

Deep neural networks (DNNs) have shown great success in many machine learning tasks. Their training is challenging since the loss surface of the network architecture is generally non-convex, or even non-smooth. How and under what assumptions is guaranteed convergence to a *global* minimum possible? We propose a reformulation of the minimization problem allowing for a new recursive algorithmic framework. By using bounded style assumptions, we prove convergence to an $\varepsilon$-(global) minimum using $\tilde{\mathcal{O}}(1/\varepsilon^3)$ gradient computations. Our theoretical foundation motivates further study, implementation, and optimization of the new algorithmic framework and further investigation of its non-standard bounded style assumptions. This new direction broadens our understanding of why and under what circumstances training of a DNN converges to a global minimum.

## 1 INTRODUCTION

In recent years, deep neural networks (DNNs) have shown a great success in many machine learning tasks. However, training these neural networks is challenging since the loss surface of network architecture is generally non-convex, or even non-smooth. Thus, there have been a long-standing question on how optimization algorithms may converge to a global minimum. Many previous work have investigated Gradient Descent algorithm and its stochastic version for over-parameterized setting (Arora et al., 2018; Soudry et al., 2018; Allen-Zhu et al., 2019; Du et al., 2019a; Zou & Gu, 2019). Although these works have shown promising convergence results under certain assumptions, there is still a lack of new efficient methods that can guarantee global convergence for machine learning optimization. In this paper, we address this problem using a different perspective. Instead of analyzing the traditional finite-sum formulation, we adopt a new *composite formulation* that exactly depicts the structure of machine learning where a data set is used to learn a common classifier.

**Representation.** Let $\left\{(x^{(i)}, y^{(i)})\right\}_{i=1}^{n}$ be a given training set with $x^{(i)} \in \mathbb{R}^m, y^{(i)} \in \mathbb{R}^c$, we investigate the following novel representation for deep learning tasks:

$$\min_{w \in \mathbb{R}^d} \left\{ F(w) = \frac{1}{n} \sum_{i=1}^{n} \phi_i(h(w; i)) \right\}, \tag{1}$$

where $h(\cdot; i) : \mathbb{R}^d \to \mathbb{R}^c, i \in [n] = \{1, \ldots, n\}$, is the classifier for each input data $x^{(i)}$; and $\phi_i : \mathbb{R}^c \to \mathbb{R}, i \in [n]$, is the loss function corresponding to each output data $y^{(i)}$. Our *composite formulation* (1) is a special case of the finite-sum problem $\min_{w \in \mathbb{R}^d} \left\{ F(w) = \frac{1}{n} \sum_{i=1}^{n} f(w; i) \right\}$ where each individual function $f(\cdot; i)$ is a composition of the loss function $\phi_i$ and the classifier $h(\cdot; i)$. This problem covers various important applications in machine learning, including logistic regression and neural networks. The most common approach for the finite-sum problem is using first-order methods such as (stochastic) gradient algorithms and making assumptions on the component functions $f(\cdot; i)$. As an alternative, we further investigate the structure of the loss function $\phi_i$ and narrow our assumption on the classifier $h(\cdot; i)$. For the purpose of this work, we first consider convex and Lipschitz-smooth loss functions while the classifiers can be non-convex. Using this representation, we propose a new framework followed by two algorithms that guarantee global convergence for the minimization problem.

**Algorithmic Framework.** Representation (1) admits a new perspective. Our key insight is to (A) define $z_i^{(t)} = h(w^{(t)}; i)$, where $t$ is an iteration count of the outer loop in our algorithmic framework.

Next (B), we want to approximate the change $z_i^{(t+1)} - z_i^{(t)}$ in terms of a step size times the gradient

$$\nabla\phi_i(z_i^{(t)}) = (\partial\phi_i(z)/\partial z_a)_{a\in[c]}\big|_{z=z_i^{(t)}},$$

and (C) we approximate the change $h(w^{(t+1)}; i) - h(w^{(t)}; i)$ in terms of the first order derivative

$$H_i^{(t)} = (\partial h_a(w; i)/\partial w_b)_{a\in[c],b\in[d]}\big|_{w=w^{(t)}}.$$

Finally, we combine (A), (B), and (C) to equate the approximations of $z_i^{(t+1)} - z_i^{(t)}$ and $h(w^{(t+1)}; i) - h(w^{(t)}; i)$. This leads to a recurrence on $w^{(t)}$ of the form $w^{(t+1)} = w^{(t)} - \eta^{(t)}v^{(t)}$, where $\eta^{(t)}$ is a step size and which involves computing $v^{(t)}$ by solving a convex quadratic subproblem, see the details in Section 4. We explain two methods for approximating a solution for the derived subproblem. We show how to approximate the subproblem by transforming it into a strongly convex problem by adding a regularizer which can be solved in closed form. And we show how to use Gradient Descent (GD) on the subproblem to find an approximation $v^{(t)}$ of its solution.

**Convergence Analysis.** Our analysis introduces non-standard bounded style assumptions. Intuitively, we assume that our convex and quadratic subproblem has a *bounded* solution. This allows us to prove a total complexity of $\tilde{\mathcal{O}}(\frac{1}{\varepsilon^3})$ to find an $\varepsilon$-(global) solution that satisfies $F(\hat{w}) - F_* \leq \varepsilon$, where $F_*$ is the global minimizer of $F$. Our analysis applies to a wide range of applications in machine learning: Our results hold for squared loss and softmax cross-entropy loss and applicable for a range of activation functions in DNN as we only assume that the $h(\cdot; i)$ are twice continuously differentiable and their Hessian matrices (second order derivatives) as well as their gradients (first order derivatives) are bounded.

**Contributions and Outline.** Our contributions in this paper can be summarized as follows.

- We propose a new representation (1) for analyzing the machine learning minimization problem. Our formulation utilizes the structure of machine learning tasks where a training data set of inputs and outputs is used to learn a common classifier. Related work in Section 2 shows how (1) is different from the classical finite-sum problem.

- Based on the new representation we propose a novel algorithm framework. The algorithmic framework approximates a solution to a subproblem for which we show two distinct approaches.

- For general DNNs and based on bounded style assumptions, we prove a total complexity of $\tilde{\mathcal{O}}(\frac{1}{\varepsilon^3})$ to find an $\varepsilon$-(global) solution that satisfies $F(\hat{w}) - F_* \leq \varepsilon$, where $F_*$ is the global minimizer of $F$.

We emphasize that our focus is on developing a new theoretical foundation and that a translation to a practical implementation with empirical results is for future work. Our theoretical foundation motivates further study, implementation, and optimization of the new algorithmic framework and further investigation of its non-standard bounded style assumptions. This new direction broadens our understanding of why and under what circumstances training of a DNN converges to a global minimum.

The rest of this paper is organized as follows. Section 2 discusses related work. Section 3 describes our setting and deep learning representation. Section 4 explains our key insight and derives our Framework 1. Section 5 presents our algorithms and their global convergence. All technical proofs are deferred to the Appendix.

## 2 RELATED WORK

**Formulation for Machine Learning Problems.** The finite-sum problem is one of the most important and fundamental problems in machine learning. Analyzing this model is the most popular approach in the machine learning literature and it has been studied intensively throughout the years (Bottou et al., 2018; Reddi et al., 2016; Duchi et al., 2011b). Our new formulation (1) is a special case of the finite-sum problem, however, it is much more complicated than the previous model since it involves the data index $i$ both inside the classifiers $h(\cdot; i)$ and the loss functions $\phi_i$. For a comparison, previous works only consider a common loss function $l(\hat{y}, y)$ for the predicted value

$\hat{y}$ and output data $y$ (Zou et al., 2018; Soudry et al., 2018). Our modified version of loss function $\phi_i$ is a natural setting for machine learning. We note that when $h(w; i)$ is the output produced by a model, our goal is to match this output with the corresponding target $y^{(i)}$. For that reason, the loss function for each output has a dependence on the output data $y^{(i)}$, and is denoted by $\phi_i$. This fact reflects the natural setting of machine learning where the outputs are designed to fit different targets, and the optimization process depends on both outer function $\phi_i$ and inner functions $h(\cdot; i)$. This complication may potentially bring a challenge to theoretical analysis. However, with separate loss functions, we believe this model will help to exploit better the structure of machine learning problems and gain more insights on the neural network architecture.

Other related composite optimization models are also investigated thoroughly in (Lewis & Wright, 2016; Zhang & Xiao, 2019; Tran-Dinh et al., 2020). Our model is different from these works as it does not have a common function wrapping outside the finite-sum term, as in (Lewis & Wright, 2016). Note that a broad class of variance reduction algorithms (e.g. SAG (Le Roux et al., 2012), SAGA (Defazio et al., 2014), SVRG (Johnson & Zhang, 2013), SARAH (Nguyen et al., 2017)) is designed specifically for the finite-sum formulation and is known to have certain benefits over Gradient Descent. In addition, the multilevel composite problem considered in (Zhang & Xiao, 2021) also covers empirical risk minimization problem. However our formulation does not match their work since our inner function $h(w; i)$ is not an independent expectation over some data distribution, but a specific function that depends on the current data.

**Global Convergence for Neural Networks.** A recent popular line of research is studying the dynamics of optimization methods on some specific neural network architectures. There are some early works that show the global convergence of Gradient Descent (GD) for simple linear network and two-layer network (Brutzkus et al., 2018; Soudry et al., 2018; Arora et al., 2019; Du et al., 2019b). Some further works extend these results to deep learning architectures (Allen-Zhu et al., 2019; Du et al., 2019a; Zou & Gu, 2019). These theoretical guarantees are generally proved for the case when the last output layer is fixed, which is not standard in practice. A recent work (Nguyen & Mondelli, 2020) prove the global convergence for GD when all layers are trained with some initial conditions. However, these results are for neural networks without bias neurons and it is unclear how these analyses can be extended to handle the bias terms of deep networks with different activations. Our novel framework and algorithms do not exclude learning bias layers as in (Nguyen & Mondelli, 2020).

Using a different algorithm, Brutzkus et al. (2018) investigate Stochastic Gradient Descent (SGD) for two-layer networks in a restricted linearly separable data setting. This line of research continues with the works from Allen-Zhu et al. (2019); Zou et al. (2018) and later with Zou & Gu (2019). They justify the global convergence of SGD for deep neural networks for some probability depending on the number of input data and the initialization process.

**Over-Paramaterized Settings and other Assumptions for Machine Learning.** Most of the modern learning architectures are over-parameterized, which means that the number of parameters are very large and often far more than the number of input data. Some recent works prove the global convergence of Gradient Descent when the number of neurons are extensively large, e.g. (Zou & Gu, 2019) requires $\Omega(n^8)$ neurons for every hidden layer, and (Nguyen & Mondelli, 2020) improves this number to $\Omega(n^3)$. If the initial point satisfies some special conditions, then they can show a better dependence of $\Omega(n)$. In Allen-Zhu et al. (2019), the authors initialize the weights using a random Gaussian distribution where the variance depends on the dimension of the problem. In non-convex setting, they prove the convergence of SGD using the assumption that the dimension depends inversely on the tolerance $\epsilon$. We will discuss how these over-paramaterized settings might be a necessary condition to develop our theory.

Other standard assumptions for machine learning include the bounded gradient assumption (Nemirovski et al., 2009; Shalev-Shwartz et al., 2007; Reddi et al., 2016; Tran et al., 2021). It is also common to assume all the iterations of an algorithm stays in a bounded domain (Duchi et al., 2011a; Levy et al., 2018; Gürbüzbalaban et al., 2019; Reddi et al., 2018; Vaswani et al., 2021). Since we are analyzing a new *composite formulation*, it is understandable that our assumptions may also not be standard. However, we believe that there is a strong connection between our assumptions and the traditional setting of machine learning. We will discuss this point more clearly in Section 4.

## 3 BACKGROUND

In this section, we discuss our formulation and notations in detail. Although this paper focuses on deep neural networks, our framework and theoretical analysis are general and applicable for other learning architectures.

**Deep Learning Representation.** Let $\{(x^{(i)}, y^{(i)})\}_{i=1}^{n}$ be a training data set where $x^{(i)} \in \mathbb{R}^m$ is a training input and $y^{(i)} \in \mathbb{R}^c$ is a training output. We consider a fully-connected neural network with $L$ layers, where the $l$-th layer, $l \in \{0, 1, \dots, L\}$, has $n_l$ neurons. We represent layer 0-th and $L$-th layer as input and output layers, respectively, that is, $n_0 = d$ and $n_L = c$. For $l \in \{1, \dots, L\}$, let $W^{(l)} \in \mathbb{R}^{n_{l-1} \times n_l}$ and $b^{(l)} \in \mathbb{R}^{n_l}$, where $\{(W^{(l)}, b^{(l)})_{l=1}^{L}\}$ represent the parameters of the neural network. A classifier $h(w; i)$ is formulated as

$$h(w; i) = W^{(L)\top} \sigma_{L-1}(W^{(L-1)\top} \sigma_{L-2}(\dots \sigma_1(W^{(1)\top} x^{(i)} + b^{(1)}) \dots) + b^{(L-1)}) + b^{(L)},$$

where $w = \mathbf{vec}(\{W^{(1)}, b^{(1)}, \dots, W^{(L)}, b^{(L)}\}) \in \mathbb{R}^d$ is the vectorized weight and $\{\sigma_l\}_{l=1}^{L-1}$ are some activation functions. The most common choices for machine learning are ReLU, sigmoid, hyperbolic tangent and softplus. For $j \in [c]$, $h_j(\cdot; i) : \mathbb{R}^d \to \mathbb{R}$ denotes the component function of the output $h(\cdot; i)$, for each data $i \in [n]$ respectively. Moreover, we define $h_i^* = \arg\min_{z \in \mathbb{R}^c} \phi_i(z), i \in [n]$.

**Loss Functions.** The well-known loss functions in neural networks for solving classification and regression problems are *softmax cross-entropy loss* and *square loss*, respectively:

*(Softmax) Cross-Entropy Loss:* $F(w) = \frac{1}{n} \sum_{i=1}^{n} f(w; i)$ with

$$f(w; i) = -y^{(i)\top} \log(\mathrm{softmax}(h(w; i))). \tag{2}$$

*Squared Loss*: $F(w) = \frac{1}{n} \sum_{i=1}^{n} f(w; i)$ with

$$f(w; i) = \frac{1}{2} \|h(w; i) - y^{(i)}\|^2. \tag{3}$$

We provide some basic definitions in optimization theory to support our theory.

**Definition 1** (*L-smooth*). *Function $\phi : \mathbb{R}^c \to \mathbb{R}$ is $L_\phi$-smooth if there exists a constant $L_\phi > 0$ such that, $\forall x_1, x_2 \in \mathbb{R}^c$,*

$$\|\nabla\phi(x_1) - \nabla\phi(x_2)\| \le L_\phi \|x_1 - x_2\|. \tag{4}$$

**Definition 2** (Convex). *Function $\phi : \mathbb{R}^c \to \mathbb{R}$ is convex if $\forall x_1, x_2 \in \mathbb{R}^c$,*

$$\phi(x_1) - \phi(x_2) \ge \langle \nabla\phi(x_2), x_1 - x_2 \rangle. \tag{5}$$

The following corollary shows the properties of softmax cross-entropy loss (2) and squared loss (3).

**Corollary 1.** *For softmax cross-entropy loss (2) and squared loss (3), there exist functions $h(\cdot; i) : \mathbb{R}^d \to \mathbb{R}^c$ and $\phi_i : \mathbb{R}^c \to \mathbb{R}$ such that, for $i \in [n]$, $\phi_i(z)$ is convex and $L_\phi$-smooth with $L_\phi = 1$, and*

$$f(w; i) = \phi_i(h(w; i)) = \phi_i(z)\big|_{z=h(w;i)}. \tag{6}$$

## 4 NEW ALGORITHM FRAMEWORK

### 4.1 KEY INSIGHT

We assume $f(w; i) = \phi_i(h(w; i))$ with $\phi_i$ convex and $L_\phi$-smooth. Our goal is to utilize the convexity of the outer function $\phi_i$. In order to simplify notation, we write $\nabla_z \phi_i(h(w^{(t)}; i))$ instead of $\nabla_z \phi_i(z)\big|_{z=h(w^{(t)};i)}$ and denote $z_i^{(t)} = h(w^{(t)}; i)$. Starting from the current weight $w^{(t)}$, we would like to find the next point $w^{(t+1)}$ that satisfies the following approximation *for all $i \in [n]$*:

$$h(w^{(t+1)}; i) = z_i^{(t+1)} \approx z_i^{(t)} - \alpha_i^{(t)} \nabla_z \phi_i(z_i^{(t)}) = h(w^{(t)}; i) - \alpha_i^{(t)} \nabla_z \phi_i(h(w^{(t)}; i)). \tag{7}$$

We can see that this approximation is a "noisy" version of a gradient descent update for every function $\phi_i$, simultaneously for all $i \in [n]$. In order to do this, we use the following update

$$w^{(t+1)} = w^{(t)} - \eta^{(t)} v^{(t)}, \tag{8}$$

where $\eta^{(t)} > 0$ is a learning rate and $v^{(t)}$ is a search direction that helps us approximate equation (7). If the update term $\eta^{(t)} v^{(t)}$ is small enough, and if $h(\cdot; i)$ has some nice smooth properties, then from basic calculus we have the following approximation:

$$h(w^{(t+1)}; i) = h(w^{(t)} - \eta^{(t)} v^{(t)}; i) \approx h(w^{(t)}; i) - H_i^{(t)}\big(\eta^{(t)} v^{(t)}\big), \tag{9}$$

where $H_i^{(t)}$ is a matrix in $\mathbb{R}^{c \times d}$ with first-order derivatives. Motivated by approximations (7) and (9), we consider the following optimization problem:

$$v_*^{(t)} = \arg\min_{v \in \mathbb{R}^d} \frac{1}{2} \frac{1}{n} \sum_{i=1}^{n} \| H_i^{(t)}\big(\eta^{(t)} v\big) - \alpha_i^{(t)} \nabla_z \phi_i(h(w^{(t)}; i)) \|^2. \tag{10}$$

Hence, by solving for the solution $v_*^{(t)}$ of problem (10) we are able to find a search direction for the key approximation (7). This yields our new algorithmic Framework 1, see below.

---

**Framework 1** New Algorithm Framework

---

**Initialization:** Choose an initial point $w^{(0)} \in \mathbb{R}^d$;
**for** $t = 0, 1, \cdots, T - 1$ **do**
  Solve for an approximation $v^{(t)}$ of the solution $v_*^{(t)}$ of the problem in (10)

$$v_*^{(t)} = \arg\min_{v \in \mathbb{R}^d} \frac{1}{2} \frac{1}{n} \sum_{i=1}^{n} \| \eta^{(t)} H_i^{(t)} v - \alpha_i^{(t)} \nabla_z \phi_i(h(w^{(t)}; i)) \|^2$$

  Update $w^{(t+1)} = w^{(t)} - \eta^{(t)} v^{(t)}$
**end for**

---

### 4.2 Technical Assumptions

**Assumption 1.** *The loss function $\phi_i$ is convex and $L_\phi$-smooth for $i \in [n]$. Moreover, we assume that it is lower bounded, i.e. $\inf_{z \in \mathbb{R}^c} \phi_i(z) > -\infty$ for $i \in [n]$.*

We have shown the convexity and smoothness of squared loss and softmax cross-entropy loss in Section 3. The bounded property of $\phi_i$ is required in any algorithm for the well-definedness of (1). Now, in order to use the Taylor series approximation, we need the following assumption on the neural network architecture $h$:

**Assumption 2.** *We assume that $h(\cdot; i)$ is twice continuously differentiable for all $i \in [n]$ (i.e. the second-order partial derivatives of all scalars $h_j(\cdot; i)$ are continuous for all $j \in [c]$ and $i \in [n]$), and that their Hessian matrices are bounded, that is, there exists a $G > 0$ such that for all $w \in \mathbb{R}^d$, $i \in [n]$ and $j \in [c]$,*

$$\|M_{i,j}(w)\| = \|\boldsymbol{J}_w\left(\nabla_w h_j(w; i)\right)\| \leq G, \tag{11}$$

*where $\boldsymbol{J}_w$ denotes the Jacobian[1].*

**Remark 1** (Relation to second-order methods). *Although our analysis requires an assumption on the Hessian matrices of $h(w; i)$, our algorithms do not use any second order information or try to approximate this information. Our theoretical analysis focused on the approximation of the classifier and the gradient information, therefore is not related to the second order type algorithms. It is currently unclear how to apply second order methods into our problem, however, this is an interesting research question to expand the scope of this work.*

---

[1] For a continuously differentiable function $g(w) : \mathbb{R}^d \to \mathbb{R}^c$ we define the Jacobian $\mathbf{J}_w(g(w))$ as the matrix $(\partial g_a(w)/\partial w_b)_{a \in [c], b \in [d]}$.

Assumption 2 allows us to apply a Taylor approximation of each function $h_j(\cdot; i)$ with which we prove the following Lemma that bounds the error in equation (9):

**Lemma 1.** *Suppose that Assumption 2 holds for the classifier $h$. Then for all $i \in [n]$ and $0 \leq t < T$,*

$$h(w^{(t+1)}; i) = h(w^{(t)} - \eta^{(t)} v^{(t)}; i) = h(w^{(t)}; i) - \eta^{(t)} H_i^{(t)} v^{(t)} + \epsilon_i^{(t)}, \tag{12}$$

*where*

$$H_i^{(t)} = \boldsymbol{J}_w(h(w; i))|_{w=w^{(t)}} \in \mathbb{R}^{c \times d} \tag{13}$$

*is defined as the Jacobian matrix of $h(w; i)$ at $w^{(t)}$ and entries $\epsilon_{i,j}^{(t)}$, $j \in [c]$, of vector $\epsilon_i^{(t)}$ satisfy*

$$|\epsilon_{i,j}^{(t)}| \leq \frac{1}{2}(\eta^{(t)})^2 \|v^{(t)}\|^2 G. \tag{14}$$

In order to approximate (7) combined with (9), that is, to make sure the right hand sides of (7) and (9) are close to one another, we consider the optimization problem (10):

$$v_*^{(t)} = \arg\min_{v \in \mathbb{R}^d} \frac{1}{2} \frac{1}{n} \sum_{i=1}^n \|\eta^{(t)} H_i^{(t)} v - \alpha_i^{(t)} \nabla_z \phi_i(h(w^{(t)}; i))\|^2.$$

The optimal value of problem (10) is equal to 0 if there exists a vector $v_*^{(t)}$ satisfying $\eta^{(t)} H_i^{(t)} v_*^{(t)} = \alpha_i^{(t)} \nabla_z \phi_i(h(w^{(t)}; i))$ for every $i \in [n]$. Since the solution $v_*^{(t)}$ is in $\mathbb{R}^d$ and $\nabla_z \phi_i(h(w^{(t)}; i))$ is in $\mathbb{R}^c$, this condition is equivalent to a linear system with $n \cdot c$ constraints and $d$ variables. In the over-parameterized setting where dimension $d$ is sufficiently large ($d \gg n \cdot c$) and there are no identical data, there exists almost surely a vector $v_*^{(t)}$ that interpolates all the training set, see the Appendix for details.

Let us note that an approximation of $v_*^{(t)}$ serves as the search direction for Framework 1. For this reason, the solution $v_*^{(t)}$ of problem (10) plays a similar role as a gradient in the search direction of (stochastic) gradient descent method. It is standard to assume a bounded gradient in the machine learning literature (Nemirovski et al., 2009; Shalev-Shwartz et al., 2007; Reddi et al., 2016). Motivated by these facts, we assume the following Assumption 3, which implies the existence of a near-optimal *bounded* solution of (10):

**Assumption 3.** *We consider an over-parameterized setting where dimension $d$ is sufficiently large enough to interpolate all the data and the tolerance $\varepsilon$. We assume that there exists a bound $V > 0$ such that for $\varepsilon > 0$ and $0 \leq t < T$ as in Framework 1, there exists a vector $\hat{v}_{*\varepsilon}^{(t)}$ with $\|\hat{v}_{*\varepsilon}^{(t)}\|^2 \leq V$ so that*

$$\frac{1}{2} \frac{1}{n} \sum_{i=1}^n \|\eta^{(t)} H_i^{(t)} \hat{v}_{*\varepsilon}^{(t)} - \alpha_i^{(t)} \nabla_z \phi_i(h(w^{(t)}; i))\|^2 \leq \varepsilon^2.$$

Our Assumption 3 requires a nice dependency on the tolerance $\varepsilon$ for the gradient matrices $H_i^{(t)}$ and $\nabla_z \phi_i(h(w^{(t)}; i))$. We note that at the starting point $t = 0$, these matrices may depend on $\varepsilon$ due to the initialization process and the dependence of $d$ on $\varepsilon$. This setting is similar to previous works, e.g. Allen-Zhu et al. (2019).

## 5 New Algorithms and Convergence Results

### 5.1 Approximating the solution using regularizer

Since problem (10) is convex and quadratic, we consider the following regularized problem:

$$\min_{v \in \mathbb{R}^d} \left\{ \Psi(v) = \frac{1}{2} \frac{1}{n} \sum_{i=1}^n \|\eta^{(t)} H_i^{(t)} v - \alpha_i^{(t)} \nabla_z \phi_i(h(w^{(t)}; i))\|^2 + \frac{\varepsilon^2}{2} \|v\|^2 \right\}, \tag{15}$$

for some small $\varepsilon > 0$ and $t \geq 0$. It is widely known that problem (15) is strongly convex, and has a unique minimizer $v^{(t)}_{*\,\text{reg}}$. The global minimizer satisfies $\nabla_v \Psi(v^{(t)}_{*\,\text{reg}}) = 0$. We have

$$\nabla_v \Psi(v) = \frac{1}{n}\sum_{i=1}^{n}[\eta^{(t)}H_i^{(t)\top}H_i^{(t)}\eta^{(t)}v - \alpha_i^{(t)}\eta^{(t)}H_i^{(t)\top}\nabla_z\phi_i(h(w^{(t)};i))] + \varepsilon^2 \cdot v$$

$$= \left(\frac{1}{n}\sum_{i=1}^{n}\eta^{(t)}H_i^{(t)\top}H_i^{(t)}\eta^{(t)} + \varepsilon^2 I\right)v - \left(\frac{1}{n}\sum_{i=1}^{n}\alpha_i^{(t)}\eta^{(t)}H_i^{(t)\top}\nabla_z\phi_i(h(w^{(t)};i))\right).$$

Therefore,

$$v^{(t)}_{*\,\text{reg}} = \left(\frac{1}{n}\sum_{i=1}^{n}\eta^{(t)}H_i^{(t)\top}H_i^{(t)}\eta^{(t)} + \varepsilon^2 I\right)^{-1}\left(\frac{1}{n}\sum_{i=1}^{n}\alpha_i^{(t)}\eta^{(t)}H_i^{(t)\top}\nabla_z\phi_i(h(w^{(t)};i))\right). \quad (16)$$

If $\varepsilon^2$ is small enough, then $v^{(t)}_{*\,\text{reg}}$ is a close approximation of the solution $v^{(t)}_*$ for problem (10). Our first algorithm updates Framework 1 based on this approximation.

---

**Algorithm 1** Solve for the exact solution of the regularized problem

---

**Initialization:** Choose an initial point $w^{(0)} \in \mathbb{R}^d$, tolerance $\varepsilon > 0$;
**for** $t = 0, 1, \cdots, T-1$ **do**
    Update the search direction $v^{(t)}$ as the solution $v^{(t)}_{*\,\text{reg}}$ of problem in (15):

$$v^{(t)} = v^{(t)}_{*\,\text{reg}} = \left(\frac{1}{n}\sum_{i=1}^{n}\eta^{(t)}H_i^{(t)\top}H_i^{(t)}\eta^{(t)} + \varepsilon^2 I\right)^{-1}\left(\frac{1}{n}\sum_{i=1}^{n}\alpha_i^{(t)}\eta^{(t)}H_i^{(t)\top}\nabla_z\phi_i(h(w^{(t)};i))\right)$$

    Update $w^{(t+1)} = w^{(t)} - \eta^{(t)}v^{(t)}$
**end for**

---

The following Lemma shows the relation between the regularized solution $v^{(t)}_{*\,\text{reg}}$ and the optimal solution of the original convex problem $\hat{v}^{(t)}_{*\varepsilon}$.

**Lemma 2.** *For given $\varepsilon > 0$, suppose that Assumption 3 holds for bound $V > 0$. Then, for iteration $0 \leq t < T$, the optimal solution $v^{(t)}_{*\,\text{reg}}$ of problem (15) satisfies $\|v^{(t)}_{*\,\text{reg}}\|^2 \leq 2 + V$ and*

$$\frac{1}{2}\frac{1}{n}\sum_{i=1}^{n}\|\eta^{(t)}H_i^{(t)}v^{(t)}_{*\,\text{reg}} - \alpha_i^{(t)}\nabla_z\phi_i(h(w^{(t)};i))\|^2 \leq (1 + \frac{V}{2})\varepsilon^2. \quad (17)$$

Based on Lemma 2, we guarantee the global convergence of Algorithm 1 and prove our first theorem. Since it is currently expensive to solve for the exact solution of problem (15), our algorithm serves as a theoretical method to obtain the global convergence for the finite-sum minimization.

**Theorem 1.** *Let $w^{(t)}$ be generated by Algorithm 1 where we use the closed form solution for the search direction. We execute Algorithm 1 for $T = \frac{\beta}{\varepsilon}$ outer loops for some constant $\beta > 0$. We assume Assumption 1 holds. Suppose that Assumption 2 holds for $G > 0$ and Assumption 3 holds for $V > 0$. We set the step size equal to $\eta^{(t)} = D\sqrt{\varepsilon}$ for some $D > 0$ and choose a learning rate $\alpha_i^{(t)} = (1 + \varepsilon)\alpha_i^{(t-1)} = (1 + \varepsilon)^t\alpha_i^{(0)}$. Based on $\beta$, we define $\alpha_i^{(0)} = \frac{\alpha}{e^\beta L_\phi}$ with $\alpha \in (0, \frac{1}{3})$. Let $F_*$ be the global minimizer of $F$, and $h_i^* = \arg\min_{z\in\mathbb{R}^c}\phi_i(z), i \in [n]$. Then*

$$\frac{1}{T}\sum_{t=0}^{T-1}[F(w^{(t)}) - F_*] \leq \frac{e^\beta L_\phi(1 + \varepsilon)}{2(1-3\alpha)\alpha\beta}\cdot\frac{1}{n}\sum_{i=1}^{n}\|h(w^{(0)};i) - h_i^*\|^2 \cdot \varepsilon$$

$$+ \frac{e^\beta L_\phi(3\varepsilon + 2)}{8\alpha(1-3\alpha)}\left[c(4 + (V+2)GD^2)^2 + 8 + 4V\right]\cdot\varepsilon. \quad (18)$$

We note that $\beta$ is a constant for the purpose of choosing the number of iterations $T$. The analysis can be simplified by choosing $\beta = 1$ with $T = \frac{1}{\varepsilon}$. Notice that the common convergence criteria for

finding a stationary point for non-convex problems is $\frac{1}{T}\sum_{t=1}^{T}||\nabla F(w_t)||^2 \leq O(\varepsilon)$. This criteria has been widely used in the existing literature for non-convex optimization problems. Our convergence criteria $\frac{1}{T}\sum_{t=1}^{T}[F(w_t) - F_*] \leq O(\varepsilon)$ is slightly different, in order to find a global solution for non-convex problems.

Our proof for Theorem 1 is novel and insightful. It is originally motivated by the Gradient Descent update (7) and the convexity of the loss functions $\phi_i$. For this reason it may not be a surprise that Algorithm 1 can find an $\varepsilon$-global solution after $\mathcal{O}\left(\frac{1}{\varepsilon}\right)$ iterations. However, computing the exact solution in every iteration might be extremely challenging, especially when the number of samples $n$ is large. Therefore, we present a different approach to this problem in the following section.

## 5.2 APPROXIMATION USING GRADIENT DESCENT

In this section, we use Gradient Descent (GD) algorithm to solve the strongly convex problem (15). It is well-known that if $\psi(x) - \frac{\mu}{2}\|x\|^2$ is convex for $\forall x \in \mathbb{R}^c$, then $\psi(x)$ is $\mu$-strongly convex (see e.g. Nesterov (2004)). Hence $\Psi(\cdot)$ is $\varepsilon^2$-strongly convex. For each iteration $t$, we use GD to find a search direction $v^{(t)}$ which is sufficiently close to the optimal solution $v_{* \text{ reg}}^{(t)}$ in that

$$\|v^{(t)} - v_{* \text{ reg}}^{(t)}\| \leq \varepsilon. \tag{19}$$

Our Algorithm 2 is described as follows.

---

**Algorithm 2** Solve the regularized problem using Gradient Descent

---

**Initialization:** Choose an initial point $w^{(0)} \in \mathbb{R}^d$, tolerance $\varepsilon > 0$;
**for** $t = 0, 1, \cdots, T-1$ **do**
  Use Gradient Descent algorithm to solve Problem (15) and find a solution $v^{(t)}$ that satisfies

$$\|v^{(t)} - v_{* \text{ reg}}^{(t)}\| \leq \varepsilon$$

  Update $w^{(t+1)} = w^{(t)} - \eta^{(t)}v^{(t)}$
**end for**

---

Since Algorithm 2 can only approximate a solution within some $\varepsilon$-preciseness, we need a supplemental assumption for the analysis of our next Theorem 2:

**Assumption 4.** *Let $H_i^{(t)}$ be the Jacobian matrix defined in Lemma 1. We assume that there exists some constant $H > 0$ such that, for $i \in [n]$, $\varepsilon > 0$, and $0 \leq t < T$ as in Algorithm 2,*

$$\|H_i^{(t)}\| \leq \frac{H}{\sqrt{\varepsilon}}. \tag{20}$$

Assumption 4 requires a mild condition on the bounded Jacobian of $h(w; i)$, and the upper bound may depend on $\varepsilon$. This flexibility allows us to accommodate a good dependence of $\varepsilon$ for the theoretical analysis. We are now ready to present our convergence theorem for Algorithm 2.

**Theorem 2.** *Let $w^{(t)}$ be generated by Algorithm 2 where $v^{(t)}$ satisfies (19). We execute Algorithm 2 for $T = \frac{\beta}{\varepsilon}$ outer loops for some constant $\beta > 0$. We assume Assumption 1 holds. Suppose that Assumption 2 holds for $G > 0$, Assumption 3 holds for $V > 0$ and Assumption 4 holds for $H > 0$. We set the step size equal to $\eta^{(t)} = D\sqrt{\varepsilon}$ for some $D > 0$ and choose a learning rate $\alpha_i^{(t)} = (1+\varepsilon)\alpha_i^{(t-1)} = (1+\varepsilon)^t\alpha_i^{(0)}$. Based on $\beta$, we define $\alpha_i^{(0)} = \frac{\alpha}{e^{\beta}L_\phi}$ with $\alpha \in (0, \frac{1}{4})$. Let $F_*$ be the global minimizer of $F$, and $h_i^* = \arg\min_{z \in \mathbb{R}^c} \phi_i(z), i \in [n]$. Then*

$$\frac{1}{T}\sum_{t=0}^{T-1}[F(w^{(t)}) - F_*] \leq \frac{e^{\beta}L_\phi(1+\varepsilon)}{2(1-4\alpha)\alpha\beta} \cdot \frac{1}{n}\sum_{i=1}^{n}\|h(w^{(0)}; i) - h_i^*\|^2 \cdot \varepsilon$$

$$+ \frac{e^{\beta}L_\phi(4\varepsilon + 3)}{2\alpha(1-4\alpha)}\left[D^2H^2 + c(2 + (V + \varepsilon^2 + 2)GD^2)^2 + 2 + V\right] \cdot \varepsilon.$$

Theorem 2 implies Corollary 2 which provides the computational complexity for Algorithm 2. Note that for (Stochastic) Gradient Descent, we derive the complexity in terms of component gradient calculations for the finite-sum problem (1). As an alternative, for Algorithm 2 we compare the number of component gradients in problem (15). Such individual gradient has the following form:

$$\nabla_v \psi_i(v) = \eta^{(t)} H_i^{(t)\top} H_i^{(t)} \eta^{(t)} v - \alpha_i^{(t)} \eta^{(t)} H_i^{(t)\top} \nabla_z \phi_i(h(w^{(t)}; i)).$$

In machine learning applications, the gradient of $f(\cdot; i)$ is calculated using automatic differentiation (i.e. backpropagation). Since $f(\cdot; i)$ is the composition of the network structure $h(\cdot; i)$ and loss function $\phi_i(\cdot)$, this process also computes the Jacobian matrix $H_i^{(t)}$ and the gradient $\nabla_z \phi_i(h(w^{(t)}; i))$ at a specific weight $w^{(t)}$. Since matrix-vector multiplication computation is not expensive, the cost for computing the component gradient of problem (15) is similar to problem (1).

**Corollary 2.** *Suppose that the conditions in Theorem 2 hold with $\eta^{(t)} = \frac{D\sqrt{\hat{\varepsilon}}}{\sqrt{N}}$ for some $D > 0$ and $0 < \hat{\varepsilon} \leq N$ (that is, we set $\varepsilon = \hat{\varepsilon}/N$), where*

$$N = \frac{e^\beta L_\phi \sum_{i=1}^n \|h(w^{(0)}; i) - h_i^*\|^2}{n(1 - 4\alpha)\alpha\beta} + \frac{7 e^\beta L_\phi \left[D^2 H^2 + c(2 + (V+3)GD^2)^2 + 2 + V\right]}{2\alpha(1 - 4\alpha)}.$$

*Then, the total complexity to guarantee $\min_{0 \leq t \leq T-1}[F(w^{(t)}) - F_*] \leq \frac{1}{T} \sum_{t=0}^{T-1}[F(w^{(t)}) - F_*] \leq \hat{\varepsilon}$ is $\mathcal{O}\left(n\frac{N^3\beta}{\hat{\varepsilon}^3}(D^2 H^2 + (\hat{\varepsilon}^2/N)) \log(\frac{N}{\hat{\varepsilon}})\right).$*

**Remark 2.** *Corollary 2 shows that $\mathcal{O}(1/\hat{\varepsilon})$ outer loop iterations are needed in order to reach an $\hat{\varepsilon}$-global solution, and it proves that each iteration needs the equivalent of $\mathcal{O}\left(\frac{n}{\hat{\varepsilon}^2} \log(\frac{1}{\hat{\varepsilon}})\right)$ gradient computations for computing an approximate solution. In total, Algorithm 2 has total complexity $\mathcal{O}\left(\frac{n}{\hat{\varepsilon}^3} \log(\frac{1}{\hat{\varepsilon}})\right)$ for finding an $\hat{\varepsilon}$-global solution.*

*For a comparison, Stochastic Gradient Descent uses a total of $\mathcal{O}(\frac{1}{\varepsilon^2})$ gradient computations to find a stationary point satisfying $\mathbb{E}[\|\nabla F(\hat{w})\|^2] \leq \varepsilon$ for non-convex problems (Ghadimi & Lan, 2013). Gradient Descent has a better complexity in terms of $\varepsilon$, i.e. $\mathcal{O}(\frac{n}{\varepsilon})$ such that $\|\nabla F(\hat{w})\|^2 \leq \varepsilon$ (Nesterov, 2004). However, both methods may not be able to reach a global solution of (1). In order to guarantee global convergence for nonconvex settings, one may resort to use Polyak-Lojasiewicz (PL) inequality (Karimi et al., 2016; Gower et al., 2021). This assumption is widely known to be strong, which implies that every stationary point is also a global minimizer.*

## 6 FURTHER DISCUSSION AND CONCLUSIONS

This paper presents an alternative *composite formulation* for solving the finite-sum optimization problem. Our formulation allows a new way of exploiting the structure of machine learning problems and the convexity of squared loss and softmax cross entropy loss, and leads to a novel algorithmic framework that guarantees global convergence (when the outer loss functions are convex and Lipschitz-smooth). Our analysis is general and can be applied to various different learning architectures, in particular, our analysis and assumptions match practical neural networks; in recent years, there has been a great interest in the structure of deep learning architectures for over-parameterized settings (Arora et al., 2018; Allen-Zhu et al., 2019; Nguyen & Mondelli, 2020). Algorithm 2 demonstrates a gradient method to solve the regularized problem, however, other methods can be applied to our framework (e.g. conjugate gradient descent).

Our theoretical foundation motivates further study, implementation, and optimization of the new algorithmic framework and further investigation of its non-standard bounded style assumptions. Possible research directions include more practical algorithm designs based on our Framework 1, and different related methods to solve the regularized problem and approximate the solution. This potentially leads to a new class of efficient algorithms for machine learning problems. This paper presents a new perspective to the research community.

ETHICS STATEMENT

This paper does not contain ethics concerns.

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

# APPENDIX

## A  TABLE OF NOTATIONS

| Notation | Meaning |
|---|---|
| $F_*$ | Global minimization function of $F$ in (1) 
 $F_* = \min_{w \in \mathbb{R}^d} F(w)$ |
| $h_i^*$ | $h_i^* = \arg\min_{z \in \mathbb{R}^c} \phi_i(z), i \in [n]$ |
| $v_*^{(t)}$ | Solution of the convex problem in (10) 
 $\min_{v \in \mathbb{R}^d} \frac{1}{2} \frac{1}{n} \sum_{i=1}^{n} \|\eta^{(t)} H_i^{(t)} v - \alpha_i^{(t)} \nabla_z \phi_i(h(w^{(t)}; i))\|^2$ |
| $v^{(t)}$ | An approximation of $v_*^{(t)}$ which is used as the search direction in Framework 1 |
| $\hat{v}_{*\varepsilon}^{(t)}$ | A vector that satisfies 
 $\frac{1}{2} \frac{1}{n} \sum_{i=1}^{n} \|\eta^{(t)} H_i^{(t)} v - \alpha_i^{(t)} \nabla_z \phi_i(h(w^{(t)}; i))\|^2 \leq \varepsilon^2$ 
 for some $\varepsilon > 0$ and $\|\hat{v}_{*\varepsilon}^{(t)}\|^2 \leq V$, for some $V > 0$. |
| $v_{*\ \text{reg}}^{(t)}$ | Solution of the strongly convex problem in (15) 
 $\min_{v \in \mathbb{R}^d} \left\{ \frac{1}{2} \frac{1}{n} \sum_{i=1}^{n} \|\eta^{(t)} H_i^{(t)} v - \alpha_i^{(t)} \nabla_z \phi_i(h(w^{(t)}; i))\|^2 + \frac{\varepsilon^2}{2} \|v\|^2 \right\}$ |

## B  USEFUL RESULTS

The following lemmas provide key tools for our results.

**Lemma 3** (Squared loss). *Let $b \in \mathbb{R}^c$ and define $\phi(z) = \frac{1}{2}\|z - b\|^2$ for $z \in \mathbb{R}^c$. Then $\phi$ is convex and $L_\phi$-smooth with $L_\phi = 1$.*

**Lemma 4** (Softmax cross-entropy loss). *Let index $a \in [c]$ and define*

$$\phi(z) = \log \left[ \sum_{k=1}^{c} \exp(z_k - z_a) \right] = \log \left[ \sum_{k=1}^{c} \exp(w_k^\top z) \right],$$

*for $z = (z_1, \ldots, z_c)^\top \in \mathbb{R}^c$, where $w_k = e_k - e_a$ with $e_i$ representing the $i$-th unit vector (containing 1 at the $i$-th position and 0 elsewhere). Then $\phi$ is convex and $L_\phi$-smooth with $L_\phi = 1$.*

The following lemma is a standard result in (Nesterov, 2004).

**Lemma 5** ((Nesterov, 2004)). *If $\phi$ is $L_\phi$-smooth and convex, then for $\forall z \in \mathbb{R}^c$,*

$$\|\nabla\phi(z)\|^2 \leq 2L_\phi(\phi(z) - \phi(z_*)), \tag{21}$$

*where $z_* = \arg\min_z \phi(z)$.*

The following useful derivations could be used later in our theoretical analysis. Since $\phi_i$ is convex, by Definition 2 we have

$$\phi_i(h(w; i)) \geq \phi_i(h(w'; i)) + \left\langle \nabla_z \phi_i(z) \Big|_{z=h(w'; i)}, h(w; i) - h(w'; i) \right\rangle. \tag{22}$$

If $\phi_i$ is convex and $L_\phi$-smooth, then by Lemma 5

$$\left\| \nabla_z \phi_i(z) \Big|_{z=h(w; i)} \right\|^2 \leq 2L_\phi \left[ \phi_i(h(w; i)) - \phi_i(h_i^*) \right], \tag{23}$$

where $h_i^* = \arg\min_{z \in \mathbb{R}^c} \phi_i(z)$.

We compute gradients of $f(w; i)$ in term of $\phi_i(h(w; i))$.

- **Gradient of softmax cross-entropy loss:**

$$\nabla \phi_i(z)\big|_{z=h(w;i)} = \left(\frac{\partial \phi_i(z)}{\partial z_1}\bigg|_{z=h(w;i)}, \ldots, \frac{\partial \phi_i(z)}{\partial z_c}\bigg|_{z=h(w;i)}\right)^\top,$$

where for $j \in [c]$,

$$\frac{\partial \phi_i(z)}{\partial z_j}\bigg|_{z=h(w;i)} = \begin{cases} \frac{\exp\left([h(w;i)]_j - [h(w;i)]_{I(y^{(i)})}\right)}{\sum_{k=1}^c \exp\left([h(w;i)]_k - [h(w;i)]_{I(y^{(i)})}\right)} & , j \neq I(y^{(i)}) \\ -\frac{\sum_{k \neq I(y^{(i)})} \exp\left([h(w;i)]_k - [h(w;i)]_{I(y^{(i)})}\right)}{\sum_{k=1}^c \exp\left([h(w;i)]_k - [h(w;i)]_{I(y^{(i)})}\right)} & , j = I(y^{(i)}) \end{cases}. \tag{24}$$

- **Gradient of squared loss:**

$$\nabla \phi_i(z)\big|_{z=h(w;i)} = h(w;i) - y^{(i)}. \tag{25}$$

## C   ADDITIONAL DISCUSSION

### C.1   ABOUT ASSUMPTION 2

We make a formal assumption for the case $h(\cdot; i)$ is closely approximated by $k(\cdot; i)$.

**Assumption 5.** *We assume that for all $i \in [n]$ there exists some approximations $k(w; i) : \mathbb{R}^d \to \mathbb{R}^c$ such that*

$$|k_j(w;i) - h_j(w;i)| \leq \varepsilon, \ \forall w \in \mathbb{R}^d, \ i \in [n] \text{ and } j \in [c], \tag{26}$$

*where $k(\cdot; i)$ are twice continuously differentiable (i.e. the second-order partial derivatives of all scalars $k_j(\cdot; i)$ are continuous for all $i \in [n]$), and that their Hessian matrices are bounded:*

$$\|M_{i,j}(w)\| = \|\boldsymbol{J}_w\left(\nabla_w k_j(w;i)\right)\| \leq G, \ \forall w \in \mathbb{R}^d, \ i \in [n] \text{ and } j \in [c]. \tag{27}$$

Assumption 5 allows us to prove the following Lemma that bound the error in equation (9):

**Lemma 6.** *Suppose that Assumption 5 holds for the classifier $h$. Then for all $i \in [n]$ and $0 \leq t < T$, we have:*

$$h(w^{(t+1)}; i) = h(w^{(t)} - \eta^{(t)} v^{(t)}; i) = h(w^{(t)}; i) - \eta^{(t)} H_i^{(t)} v^{(t)} + \epsilon_i^{(t)}, \tag{28}$$

*where $H_i^{(t)}$ is defined to be the Jacobian matrix of the approximation $k(w; i)$ at $w^{(t)}$:*

$$H_i^{(t)} := \boldsymbol{J}_w k(w;i)|_{w=w^{(t)}} = \begin{bmatrix} \frac{\partial k_1(w;i)}{\partial w_1} & \cdots & \frac{\partial k_1(w;i)}{\partial w_d} \\ \cdots & \cdots & \cdots \\ \frac{\partial k_c(w;i)}{\partial w_1} & \cdots & \frac{\partial k_c(w;i)}{\partial w_d} \end{bmatrix}\Bigg|_{w=w^{(t)}} \in \mathbb{R}^{c \times d}. \tag{29}$$

*Additionally we have,*

$$|\epsilon_{i,j}^{(t)}| \leq \frac{1}{2}(\eta^{(t)})^2 \|v^{(t)}\|^2 G + 2\varepsilon, \ j \in [c]. \tag{30}$$

Note that these result recover the case when $h(\cdot; i)$ is itself smooth. Hence we analyze our algorithms using the result of Lemma 6, which generalizes the result from Lemma 1.

### C.2   ABOUT ASSUMPTION 3

In this section, we justify the existence of the search direction in Assumption 3 (almost surely). We argue that there exists a vector $\hat{v}_{*\varepsilon}^{(t)}$ satisfying

$$\frac{1}{2}\frac{1}{n}\sum_{i=1}^n \|\eta^{(t)} H_i^{(t)} \hat{v}_{*\varepsilon}^{(t)} - \alpha_i^{(t)} \nabla_z \phi_i(h(w^{(t)};i))\|^2 \leq \varepsilon^2.$$

It is sufficient to find a vector $v$ satisfying that

$$\eta^{(t)} H_i^{(t)} v = \alpha_i^{(t)} \nabla_z \phi_i(h(w^{(t)}; i)) \text{ for every } i \in [n].$$

Since the solution $v$ is in $\mathbb{R}^d$ and $\nabla_z \phi_i(h(w^{(t)}; i))$ is in $\mathbb{R}^c$, this condition is equivalent to a linear system with $n \cdot c$ constraints and $d$ variables. Let $A$ and $b$ be the following stacked matrix and vector:

$$A = \begin{bmatrix} H_1^{(t)} \eta^{(t)} \\ \ldots \\ H_n^{(t)} \eta^{(t)} \end{bmatrix} \in \mathbb{R}^{n \cdot c \times d}, \text{ and } b = \begin{bmatrix} \alpha_1^{(t)} \nabla_z \phi_1(h(w^{(t)}; i)) \\ \ldots \\ \alpha_n^{(t)} \nabla_z \phi_n(h(w^{(t)}; i)) \end{bmatrix} \in \mathbb{R}^{n \cdot c},$$

then the problem reduce to finding the solution of the equation $Av = b$. In the over-parameterized setting where dimension $d$ is sufficiently large ($d \gg n \cdot c$), then rank $A = n \cdot c$ almost surely and there exists almost surely a vector $v$ that interpolates all the training set.

To demonstrate this fact easier, we consider a simple neural network where the classifier $h(w; i)$ is formulated as

$$h(w; i) = W^{(2)\top} \sigma(W^{(1)\top} x^{(i)}),$$

where $c = 1$, $W^{(1)} \in \mathbb{R}^{m \times l}$ and $W^{(2)} \in \mathbb{R}^{l \times 1}$, $w = \mathbf{vec}(\{W^{(1)}, W^{(2)}\}) \in \mathbb{R}^d$ is the vectorized weight where $d = l(m + 1)$ and $\sigma$ is sigmoid activation function.

$H_i^{(t)}$ is defined to be the Jacobian matrix of $h(w; i)$ at $w^{(t)}$:

$$H_i^{(t)} := \mathbf{J}_w h(w; i)|_{w=w^{(t)}} = \begin{bmatrix} \frac{\partial h(w; i)}{\partial w_1} & \ldots & \frac{\partial h(w; i)}{\partial w_d} \end{bmatrix}\bigg|_{w=w^{(t)}} \in \mathbb{R}^{1 \times d},$$

then

$$A = \eta^{(t)} \begin{bmatrix} H_1^{(t)} \\ \ldots \\ H_n^{(t)} \end{bmatrix} = \eta^{(t)} \begin{bmatrix} \frac{\partial h(w;1)}{\partial w_1} & \ldots & \frac{\partial h(w;1)}{\partial w_d} \\ \ldots & \ldots & \ldots \\ \frac{\partial h(w;n)}{\partial w_1} & \ldots & \frac{\partial h(w;n)}{\partial w_d} \end{bmatrix} \in \mathbb{R}^{n \times d}.$$

We want to show that $A$ has full rank, almost surely. We consider the over-parameterized setting where the last layer has at least $n$ neuron (i.e. $l = n$ and the simple version when $c = 1$. We argue that rank of matrix $A$ is greater than or equal to rank of the submatrix $B$ created by the weights of the last layer $W^{(2)} \in \mathbb{R}^n$:

$$B = \begin{bmatrix} \frac{\partial h(w;1)}{\partial W_1^{(2)}} & \ldots & \frac{\partial h(w;1)}{\partial W_n^{(2)}} \\ \ldots & \ldots & \ldots \\ \frac{\partial h(w;n)}{\partial W_1^{(2)}} & \ldots & \frac{\partial h_1(w;n)}{\partial W_n^{(2)}} \end{bmatrix} \in \mathbb{R}^{n \times n}.$$

Note that $h(\cdot, i)$ is a linear function of the last weight layers (in this simple case $W^{(2)} \in \mathbb{R}^n$ and $\sigma(W^{(1)\top} x^{(i)}) \in \mathbb{R}^n$), we can compute the partial derivatives as follows:

$$\frac{\partial h(w; i)}{\partial W^{(2)}} = \sigma(W^{(1)\top} x^{(i)}); \ i \in [n].$$

Hence

$$B = \begin{bmatrix} \sigma(W^{(1)\top} x^{(1)}) \\ \ldots \\ \sigma(W^{(1)\top} x^{(n)}) \end{bmatrix} \in \mathbb{R}^{n \times n}.$$

Assuming that there are no identical data, and $\sigma$ is the sigmoid activation, the set of weights $W^{(1)}$ that make matrix $B$ degenerate has measure zero. Hence $B$ has full rank almost surely, and we have the same conclusion for $A$. Therefore we are able to prove the almost surely existence of a solution $v$ of the linear equation $Av = b$ for simple two layers network. Using the same argument, this result can be generalized for larger neural networks where the dimension $d$ is sufficiently large ($d \gg nc$).

### C.3 INITIALIZATION EXAMPLE

Our Assumption 3 requires a nice dependency on the tolerance $\varepsilon$ for the gradient matrices $H_i^{(0)}$ and $\nabla_z \phi_i(h(w^{(0)}; i))$. We note that at the starting point $t = 0$, these matrices may depend on $\varepsilon$ due to the initialization process and the dependence of $d$ on $\varepsilon$. In order to accommodate the choice of learning rate $\eta^{(0)} = D\sqrt{\varepsilon}$ in our theorems, in this section we describe a network initialization that satisfies $\|H_i^{(0)}\| = \Theta\left(\frac{1}{\sqrt{\varepsilon}}\right)$ where the gradient norm $\|\nabla_z \phi_i(h(w^{(0)}; i))\|$ is at most constant order with respect to $\varepsilon$. To simplify the problem, we only consider small-dimension data and networks without activation.

**About the target vector:** We choose $\phi_i$ to be the softmax cross-entropy loss. By Lemma 7 (see below), we have that the gradient norm is upper bounded by a constant $c$, where $c$ is the output dimension of the problem and is not dependent on $\varepsilon$. Note that when we stack all gradients for $n$ data points, then the size of new vector is still not dependent on $\varepsilon$.

**About the network architecture:** For simplicity, we consider the following classification problem where

- The input data is in $\mathbb{R}^2$. There are only two data points $\{x^{(1)}, x^{(2)}\}$. Input data is bounded and non-degenerate (we will clarify this property later).
- The output data is (categorical) in $\mathbb{R}^2$: $\{y^{(1)} = (1, 0), y^{(2)} = (0, 1)\}$.

We want to have an over-parameterized setting where the dimension of weight vector is at least $nc = 4$. We consider a simple network with two layers, no biases and no activation functions. Let the number of neurons in the hidden layer be $m$. The flow of this network is (in) $\mathbb{R}^2 \to \mathbb{R}^m \to \mathbb{R}^2$ (out). First, we consider the case where $m = 1$.

- The first layer has 2 parameters $(w_1, w_2)$ and only 1 neuron that outputs $z^{(i)} = w_1 x_1^{(i)} + w_2 x_2^{(i)}$ (the subscript is for the coordinate of input data $x^{(i)}$).
- The second layer has 2 parameters $(w_3, w_4)$. The final output is

$$h(w, i) = [w_3(w_1 x_1^{(i)} + w_2 x_2^{(i)}), w_4(w_1 x_1^{(i)} + w_2 x_2^{(i)})]^\top \in \mathbb{R}^2,$$

with $w = [w_1, w_2, w_3, w_4]^\top \in \mathbb{R}^4$. This network satisfies that the Hessian matrices of $h(w; i)$ are bounded. Let $Q$ and $b$ be the following stacked matrix and vector:

$$Q = \begin{bmatrix} H_1^{(0)} \\ H_2^{(0)} \end{bmatrix} \in \mathbb{R}^{4 \times 4}, \text{ and } b = \begin{bmatrix} \nabla_z \phi_1(h(w^{(0)}; 1)) \\ \nabla_z \phi_2(h(w^{(0)}; 2)) \end{bmatrix} \in \mathbb{R}^4,$$

Then we have the following:

$$Q = Q(w) = \begin{bmatrix} H_1^{(0)} \\ H_2^{(0)} \end{bmatrix} = \begin{bmatrix} \nabla_w[w_3(w_1 x_1^{(1)} + w_2 x_2^{(1)})] \\ \nabla_w[w_4(w_1 x_1^{(1)} + w_2 x_2^{(1)})] \\ \nabla_w[w_3(w_1 x_1^{(2)} + w_2 x_2^{(2)})] \\ \nabla_w[w_4(w_1 x_1^{(2)} + w_2 x_2^{(2)})] \end{bmatrix}$$

$$= \begin{bmatrix} w_3 x_1^{(1)} & w_3 x_2^{(1)} & w_1 x_1^{(1)} + w_2 x_2^{(1)} & 0 \\ w_4 x_1^{(1)} & w_4 x_2^{(1)} & 0 & w_1 x_1^{(1)} + w_2 x_2^{(1)} \\ w_3 x_1^{(2)} & w_3 x_2^{(2)} & w_1 x_1^{(2)} + w_2 x_2^{(2)} & 0 \\ w_4 x_1^{(2)} & w_4 x_2^{(2)} & 0 & w_1 x_1^{(2)} + w_2 x_2^{(2)} \end{bmatrix}.$$

The determinant of this matrix is a polynomial of the weight $w$ and the input data. Under some mild non-degenerate condition of the input data, we can choose some base point $w'$ that made this matrix invertible (note that if this condition is not satisfied, we can rescale/add a very small noise to the data - which is the common procedure in machine learning).

Hence the system $Qu = b$ always has a solution. Now we consider the following two initializations:

1. We choose to initialize the starting point at $w^{(0)} = \frac{1}{\sqrt{\varepsilon}}w'$ and note that $Q(w)$ is a linear function of $w$ and $Q(w')$ is independent of $\varepsilon$. Then the norm of matrix $Q(w^{(0)})$ has the same scale with $\frac{1}{\sqrt{\varepsilon}}$.

2. Instead of choosing $m = 1$, we consider an over-parameterized network where $m = \frac{1}{\varepsilon}$ (recall that $m$ is the number of neurons in the hidden layer). The hidden layer in this case is:

$$z = \begin{cases} z_1^{(i)} & = w_{1,1}^{(1)}x_1^{(i)} + w_{2,1}^{(1)}x_2^{(i)} \\ \dots \\ z_m^{(i)} & = w_{1,m}^{(1)}x_1^{(i)} + w_{2,m}^{(1)}x_2^{(i)} \end{cases}.$$

The output layer is:

$$\begin{cases} y_1^{(i)} = z_1^{(i)}w_{1,1}^{(2)} + \cdots + z_m^{(i)}w_{m,1}^{(2)} = (w_{1,1}^{(1)}x_1^{(i)} + w_{2,1}^{(1)}x_2^{(i)})w_{1,1}^{(2)} + \cdots + (w_{1,m}^{(1)}x_1^{(i)} + w_{2,m}^{(1)}x_2^{(i)})w_{m,1}^{(2)} \\ y_2^{(i)} = z_1^{(i)}w_{1,2}^{(2)} + \cdots + z_m^{(i)}w_{m,2}^{(2)} = (w_{1,1}^{(1)}x_1^{(i)} + w_{2,1}^{(1)}x_2^{(i)})w_{1,2}^{(2)} + \cdots + (w_{1,m}^{(1)}x_1^{(i)} + w_{2,m}^{(1)}x_2^{(i)})w_{m,2}^{(2)} \end{cases}$$

with $w = [w_{1,1}^{(1)}, \ldots, w_{1,m}^{(1)}, w_{2,1}^{(1)}, \ldots, w_{2,m}^{(1)}, w_{1,1}^{(2)}, w_{1,2}^{(2)}, \ldots, w_{m,1}^{(2)}, w_{m,2}^{(2)}]^\top \in \mathbb{R}^{4m}$.

Hence,

$$Q(w) = \begin{bmatrix} w_{1,1}^{(2)}x_1^{(1)} & \dots & w_{m,1}^{(2)}x_1^{(1)} & w_{1,1}^{(2)}x_2^{(1)} & \dots & w_{m,1}^{(2)}x_2^{(1)} & z_1^{(1)} & 0 & \dots z_m^{(1)} & 0 \\ w_{1,2}^{(2)}x_1^{(1)} & \dots & w_{m,2}^{(2)}x_1^{(1)} & w_{1,2}^{(2)}x_2^{(1)} & \dots & w_{m,2}^{(2)}x_2^{(1)} & 0 & z_1^{(1)} & \dots 0 & z_m^{(1)} \\ w_{1,1}^{(2)}x_1^{(2)} & \dots & w_{m,1}^{(2)}x_1^{(2)} & w_{1,1}^{(2)}x_2^{(2)} & \dots & w_{m,1}^{(2)}x_2^{(2)} & z_1^{(2)} & 0 & \dots z_m^{(2)} & 0 \\ w_{1,2}^{(2)}x_1^{(2)} & \dots & w_{m,2}^{(2)}x_1^{(2)} & w_{1,2}^{(2)}x_2^{(2)} & \dots & w_{m,2}^{(2)}x_2^{(2)} & 0 & z_1^{(2)} & \dots 0 & z_m^{(2)} \end{bmatrix}.$$

Hence, the number of (possibly) non-zero elements in each row is $3m = \frac{3}{\varepsilon}$.

For matrix $A$ of rank $r$, we have $\|A\|_2 \le \|A\|_F \le \sqrt{r}\|A\|_2$. Since the rank of $Q(w)$ is at most 4 ($nc = 4$, independent of $\varepsilon$), we only need to find the Frobenius norm of $Q(w)$. We have

$$\|Q(w)\|_F = \sqrt{\sum_{i=1}^{4}\sum_{j=1}^{4m}|q_{ij}|^2}.$$

Let $q_{min}$ and $q_{max}$ be the element with smallest/largest magnitude of $Q(w)$. Suppose that $x^{(i)} \ne (0,0)$ and choose $w \ne 0$ such that $z \ne 0$, $q_{min} > 0$ and independent of $\varepsilon$. Hence, $\frac{\sqrt{8}}{\sqrt{\varepsilon}}|q_{min}| \le \|Q(w)\|_F \le \frac{\sqrt{12}}{\sqrt{\varepsilon}}|q_{max}|$.

Hence, $\|Q(w)\| = \Theta\left(\frac{1}{\sqrt{\varepsilon}}\right)$. Therefore this simple network initialization supports the dependence on $\varepsilon$ for our Assumption 3. We note that a similar setting is found in (Allen-Zhu et al., 2019), where the authors initialize the weights using a random Gaussian distribution with a variance depending on the dimension of the problem. In non-convex setting, they prove the convergence of SGD using the assumption that the number of neurons $m$ depends inversely on the tolerance $\varepsilon$.

**Lemma 7.** *For softmax cross-entropy loss, and $x = h(w; i) \in \mathbb{R}^c$, for $\forall w \in \mathbb{R}^d$ and $i \in [n]$, we have*

$$\left\|\nabla_z \phi_i(x)\Big|_{x=h(w;i)}\right\|^2 \le c. \tag{31}$$

*Proof.* By (24), we have for $i = 1, \ldots, n$,

- For $j \neq I(y^{(i)})$:

$$
\left( \frac{\partial \phi_i(x)}{\partial x_j} \Big|_{x=h(w;i)} \right)^2 = \left( \frac{\exp\left( [h(w;i)]_j - [h(w;i)]_{I(y^{(i)})} \right)}{\sum_{k=1}^c \exp\left( [h(w;i)]_k - [h(w;i)]_{I(y^{(i)})} \right)} \right)^2
$$

$$
= \left( \frac{\exp\left( [h(w;i)]_j - [h(w;i)]_{I(y^{(i)})} \right)}{1 + \sum_{k \neq I(y^{(i)})} \exp\left( [h(w;i)]_k - [h(w;i)]_{I(y^{(i)})} \right)} \right)^2 \leq 1.
$$

- For $j = I(y^{(i)})$:

$$
\left( \frac{\partial \phi_i(x)}{\partial x_j} \Big|_{x=h(w;i)} \right)^2 = \left( \frac{\sum_{k \neq I(y^{(i)})} \exp\left( [h(w;i)]_k - [h(w;i)]_{I(y^{(i)})} \right)}{\sum_{k=1}^c \exp\left( [h(w;i)]_k - [h(w;i)]_{I(y^{(i)})} \right)} \right)^2
$$

$$
= \left( \frac{\sum_{k \neq I(y^{(i)})} \exp\left( [h(w;i)]_k - [h(w;i)]_{I(y^{(i)})} \right)}{1 + \sum_{k \neq I(y^{(i)})} \exp\left( [h(w;i)]_k - [h(w;i)]_{I(y^{(i)})} \right)} \right)^2 \leq 1
$$

Hence, for $i = 1, \dots, n$,

$$
\left\| \nabla_z \phi_i(x) \Big|_{x=h(w;i)} \right\|^2 = \sum_{j=1}^c \left( \frac{\partial \phi_i(x)}{\partial x_j} \Big|_{x=h(w;i)} \right)^2 \leq c.
$$

This completes the proof. $\qquad\qquad\square$

## D  PROOFS OF LEMMAS AND COROLLARY 1

PROOF OF LEMMA 1

*Proof.* Since $h(\cdot; i)$ are twice continuously differentiable for all $i \in [n]$, we have the following Taylor approximation for each component outputs $h_j(\cdot; i)$ where $j \in [c]$ and $i \in [n]$:

$$
h_j(w^{(t+1)}; i) = h_j(w^{(t)} - \eta^{(t)} v^{(t)}; i)
$$

$$
= h_j(w^{(t)}; i) - \mathbf{J}_w h_j(w; i)|_{w=w^{(t)}} \eta^{(t)} v^{(t)} + \frac{1}{2} (\eta^{(t)} v^{(t)})^\top M_{i,j}(\tilde{w}^{(t)})(\eta^{(t)} v^{(t)}),
$$

$$(32)$$

where $M_{i,j}(\tilde{w}^{(t)})$ is the Hessian matrices of $h_j(\cdot; i)$ at $\tilde{w}^{(t)}$ and $\tilde{w}^{(t)} = \alpha w^{(t)} + (1-\alpha) w^{(t+1)}$ for some $\alpha \in [0, 1]$. This leads to our desired statement:

$$
h(w^{(t+1)}; i) = h(w^{(t)} - \eta^{(t)} v^{(t)}; i) = h(w^{(t)}; i) - \eta^{(t)} H_i^{(t)} v^{(t)} + \epsilon_i^{(t)},
$$

where

$$
\epsilon_{i,j}^{(t)} = \frac{1}{2} (\eta^{(t)} v^{(t)})^\top M_{i,j}(\tilde{w}^{(t)})(\eta^{(t)} v^{(t)}), \ j \in [c],
$$

Hence we get the final bound:

$$
|\epsilon_{i,j}^{(t)}| \leq \frac{1}{2} \left| (\eta^{(t)} v^{(t)})^\top M_{i,j}(\tilde{w}^{(t)})(\eta^{(t)} v^{(t)}) \right|
$$

$$
\leq \frac{1}{2} (\eta^{(t)})^2 \|v^{(t)}\|^2 \cdot \|M_{i,j}(\tilde{w}^{(t)})\|
$$

$$
\overset{(11)}{\leq} \frac{1}{2} (\eta^{(t)})^2 \|v^{(t)}\|^2 G, \ j \in [c].
$$

$$\qquad\qquad\square$$

PROOF OF LEMMA 2

*Proof.* From Assumption 3, we know that there exists $\hat{v}_{*\varepsilon}^{(t)}$ so that

$$\frac{1}{2}\frac{1}{n}\sum_{i=1}^{n}\|\eta^{(t)}H_i^{(t)}\hat{v}_{*\varepsilon}^{(t)} - \alpha_i^{(t)}\nabla_z\phi_i(h(w^{(t)};i))\|^2 \leq \varepsilon^2,$$

and $\|\hat{v}_{*\varepsilon}^{(t)}\|^2 \leq V$, for some $V > 0$. Hence,

$$\frac{1}{2}\frac{1}{n}\sum_{i=1}^{n}\|\eta^{(t)}H_i^{(t)}\hat{v}_{*\varepsilon}^{(t)} - \alpha_i^{(t)}\nabla_z\phi_i(h(w^{(t)};i))\|^2 + \frac{\varepsilon^2}{2}\|\hat{v}_{*\varepsilon}^{(t)}\|^2 \leq \varepsilon^2 + \frac{\varepsilon^2}{2}V = (1 + \frac{V}{2})\varepsilon^2.$$

Since $v_{*\text{ reg}}^{(t)}$ is the optimal solution of the problem in (15) for $0 \leq t < T$, we have

$$\frac{1}{2}\frac{1}{n}\sum_{i=1}^{n}\|\eta^{(t)}H_i^{(t)}v_{*\text{ reg}}^{(t)} - \alpha_i^{(t)}\nabla_z\phi_i(h(w^{(t)};i))\|^2 + \frac{\varepsilon^2}{2}\|v_{*\text{ reg}}^{(t)}\|^2 \leq (1 + \frac{V}{2})\varepsilon^2.$$

Therefore, we have (17) and $\|v_{*\text{ reg}}^{(t)}\|^2 \leq 2 + V$ for $0 \leq t < T$. □

PROOF OF LEMMA 3

*Proof.* 1. We want to show that for any $\alpha \in [0, 1]$

$$\phi(\alpha z_1 + (1 - \alpha)z_2) \leq \alpha\phi(z_1) + (1 - \alpha)\phi(z_2), \ \forall z_1, z_2 \in \mathbb{R}^c, \tag{33}$$

in order to have the convexity of $\phi$ with respect to $z$ (see (Nesterov, 2004)).

For any $\alpha \in [0, 1]$, we have for $\forall z_1, z_2 \in \mathbb{R}^c$,

$$\begin{aligned}
&\alpha\|z_1 - b\|^2 + (1 - \alpha)\|z_2 - b\|^2 - \|\alpha(z_1 - b) + (1 - \alpha)(z_2 - b)\|^2 \\
&= \alpha\|z_1 - b\|^2 + (1 - \alpha)\|z_2 - b\|^2 - \alpha^2\|z_1 - b\|^2 - (1 - \alpha)^2\|z_2 - b\|^2 \\
&\quad - 2\alpha(1 - \alpha)\langle z_1 - b, z_2 - b\rangle \\
&\geq \alpha(1 - \alpha)\|z_1 - b\|^2 + (1 - \alpha)\alpha\|z_2 - b\|^2 - 2\alpha(1 - \alpha)\|z_1 - b\| \cdot \|z_2 - b\| \\
&= \alpha(1 - \alpha)\left(\|z_1 - b\| - \|z_2 - b\|\right)^2 \geq 0,
\end{aligned}$$

where the first inequality follows according to Cauchy-Schwarz inequality $\langle a, b\rangle \leq \|a\|\cdot\|b\|$. Hence,

$$\frac{1}{2}\|\alpha z_1 + (1 - \alpha)z_2 - b\|^2 \leq \frac{\alpha}{2}\|z_1 - b\|^2 + \frac{(1 - \alpha)}{2}\|z_2 - b\|^2.$$

Therefore, (33) implies the convexity of $\phi$ with respect to $z$.

2. We want to show that $\exists L_\phi > 0$ such that

$$\|\nabla\phi(z_1) - \nabla\phi(z_2)\| \leq L_\phi\|z_1 - z_2\|, \ \forall z_1, z_2 \in \mathbb{R}^c. \tag{34}$$

Notice that $\nabla\phi(z) = z - b$, then clearly $\forall z_1, z_2 \in \mathbb{R}^c$,

$$\|\nabla\phi(z_1) - \nabla\phi(z_2)\| = \|z_1 - z_2\|.$$

Therefore, (34) implies the $L_\phi$-smoothness of $\phi$ with respect to $z$ with $L_\phi = 1$. □

PROOF OF LEMMA 4

*Proof.* 1. For $\forall z_1, z_2 \in \mathbb{R}^c$ and $1 \leq k \leq c$, denote $u_{k,1} = \exp(w_k^\top z_1)$ and $u_{k,2} = \exp(w_k^\top z_2)$ and using Holder inequality

$$\sum_{k=1}^{c} a_k \cdot b_k \leq \left(\sum_{k=1}^{c}|a_k|^p\right)^{\frac{1}{p}}\left(\sum_{k=1}^{c}|b_k|^q\right)^{\frac{1}{q}}, \ \text{where } \frac{1}{p} + \frac{1}{q} = 1, \tag{35}$$

we have

$$
\begin{aligned}
\phi(\alpha z_1 + (1-\alpha)z_2) &= \log\left[\sum_{k=1}^{c}\exp(w_k^\top(\alpha z_1 + (1-\alpha)z_2))\right] = \log\left[\sum_{k=1}^{c}u_{k,1}^{\alpha}\cdot u_{k,2}^{(1-\alpha)}\right] \\
&\overset{(35)}{\leq} \log\left[\left(\sum_{k=1}^{c}u_{k,1}^{\alpha\cdot\frac{1}{\alpha}}\right)^{\alpha}\left(\sum_{k=1}^{c}u_{k,2}^{(1-\alpha)\cdot\frac{1}{(1-\alpha)}}\right)^{1-\alpha}\right] \\
&= \alpha\log\left[\sum_{k=1}^{c}\exp(w_k^\top z_1)\right] + (1-\alpha)\log\left[\sum_{k=1}^{c}\exp(w_k^\top z_2)\right] \\
&= \alpha\phi(z_1) + (1-\alpha)\phi(z_2),
\end{aligned}
$$

where the first inequality since $log(x)$ is an increasing function for $\forall x > 0$ and $\exp(v) > 0$ for $\forall v \in \mathbb{R}$. Therefore, (33) implies the convexity of $\phi$ with respect to $z$.

2. Note that $\|\nabla^2\phi(z)\| \leq L_\phi$ if and only if $\phi(z)$ is $L_\phi$-smooth (see (Nesterov, 2004)). First, we compute gradient of $\phi(z)$:

- For $i \neq a$:

$$
\frac{\partial\phi(z)}{\partial z_i} = \frac{\exp(z_i - z_a)}{\sum_{k=1}^{c}\exp(z_k - z_a)}.
$$

- For $i = a$:

$$
\begin{aligned}
\frac{\partial\phi(z)}{\partial z_i} &= \frac{-\sum_{k\neq a}\exp(z_k - z_a)}{\sum_{k=1}^{c}\exp(z_k - z_a)} = \frac{-\sum_{k=1}^{c}\exp(z_k - z_a) + 1}{\sum_{k=1}^{c}\exp(z_k - z_a)} \\
&= -1 + \frac{1}{\sum_{k=1}^{c}\exp(z_k - z_a)} = -1 + \frac{\exp(z_i - z_a)}{\sum_{k=1}^{c}\exp(z_k - z_a)}.
\end{aligned}
$$

We then calculate $\frac{\partial^2\phi(z)}{\partial z_j\partial z_i} = \frac{\partial}{\partial z_j}\left(\frac{\partial\phi(z)}{\partial z_i}\right)$

- For $i = j$:

$$
\begin{aligned}
\frac{\partial^2\phi(z)}{\partial z_j\partial z_i} &= \frac{\exp(z_i - z_a)[\sum_{k=1}^{c}\exp(z_k - z_a)] - \exp(z_i - z_a)\exp(z_i - z_a)}{[\sum_{k=1}^{c}\exp(z_k - z_a)]^2} \\
&= \frac{\exp(z_i - z_a)[\sum_{k=1}^{c}\exp(z_k - z_a) - \exp(z_i - z_a)]}{[\sum_{k=1}^{c}\exp(z_k - z_a)]^2}.
\end{aligned}
$$

- For $i \neq j$:

$$
\frac{\partial^2\phi(z)}{\partial z_j\partial z_i} = \frac{-\exp(z_j - z_a)\exp(z_i - z_a)}{[\sum_{k=1}^{c}\exp(z_k - z_a)]^2}.
$$

Denote that $y_i = \exp(z_i - z_a) \geq 0, i \in [c]$, we have:

- For $i = j$:

$$
\left|\frac{\partial^2\phi(z)}{\partial z_j\partial z_i}\right| = \left|\frac{y_i(\sum_{k=1}^{c}y_k - y_i)}{(\sum_{k=1}^{c}y_k)^2}\right|.
$$

- For $i \neq j$:

$$
\left|\frac{\partial^2\phi(z)}{\partial z_j\partial z_i}\right| = \frac{|y_iy_j|}{(\sum_{k=1}^{c}y_k)^2}.
$$

Recall that for matrix $A = (a_{ij}) \in \mathbb{R}^{c \times c}$: $\|A\|^2 \le \|A\|_F^2 = \sum_{i=1}^{c} \sum_{j=1}^{c} |a_{ij}|^2$. We have:

$$\sum_{j=1}^{c} \left| \frac{\partial^2 \phi(z)}{\partial z_j \partial z_i} \right|^2 \le \frac{1}{(\sum_{k=1}^{c} y_k)^4} \left[ y_i^2 (\sum_{k=1}^{c} y_k - y_i)^2 + \sum_{j \ne i} (y_i y_j)^2 \right]$$

$$= \frac{1}{(\sum_{k=1}^{c} y_k)^4} \left[ y_i^2 (\sum_{k=1}^{c} y_k)^2 - 2 y_i^2 \sum_{k=1}^{c} y_k . y_i + y_i^4 + \sum_{j \ne i} (y_i y_j)^2 \right]$$

$$= \frac{1}{(\sum_{k=1}^{c} y_k)^4} \left[ y_i^2 (\sum_{k=1}^{c} y_k)^2 - 2 y_i^3 \sum_{k=1}^{c} y_k + y_i^2 \sum_{k=1}^{c} y_k^2 \right]$$

Therefore,

$$\|\nabla^2 \phi(z)\|^2 \le \sum_{i=1}^{c} \sum_{j=1}^{c} \left| \frac{\partial^2 \phi(z)}{\partial z_j \partial z_i} \right|^2$$

$$\le \frac{1}{(\sum_{k=1}^{c} y_k)^4} \left[ (\sum_{i=1}^{c} y_i^2)(\sum_{k=1}^{c} y_k)^2 - 2(\sum_{i=1}^{c} y_i^3)(\sum_{k=1}^{c} y_k) + (\sum_{i=1}^{c} y_i^2)(\sum_{k=1}^{c} y_k^2) \right]$$

$$\le \frac{(\sum_{i=1}^{c} y_i^2)(\sum_{k=1}^{c} y_k)^2}{(\sum_{k=1}^{c} y_k)^4} \le \frac{(\sum_{k=1}^{c} y_k)^4}{(\sum_{k=1}^{c} y_k)^4} = 1,$$

where the last inequality holds since

$$(\sum_{i=1}^{c} y_i^2)(\sum_{k=1}^{c} y_k^2) \le (\sum_{i=1}^{c} y_i^3)(\sum_{k=1}^{c} y_k) \Leftrightarrow (\sum_{k=1}^{c} y_k^2) \le \sqrt{(\sum_{i=1}^{c} y_i^3)(\sum_{k=1}^{c} y_k)},$$

which follows by the application of Holder inequality (35) with $p = 2$, $q = 2$, $a_k = y_k^{3/2}$, and $b_k = y_k^{1/2}$ (Note that $y_k \ge 0$, $k \in [c]$). Hence, $\|\nabla^2 \phi(z)\| \le L_\phi$ with $L_\phi = 1$ which is equivalent to $L_\phi$-smoothness of $\phi$. $\qquad \square$

PROOF OF LEMMA 6

*Proof.* Since $k(\cdot; i)$ are twice continuously differentiable for all $i \in [n]$, we have the following Taylor approximation for each component outputs $k_j(\cdot; i)$ where $j \in [c]$ and $i \in [n]$:

$$k_j(w^{(t+1)}; i) = k_j(w^{(t)} - \eta^{(t)} v^{(t)}; i)$$
$$= k_j(w^{(t)}; i) - \mathbf{J}_w k_j(w; i)|_{w=w^{(t)}} \eta^{(t)} v^{(t)} + \frac{1}{2} (\eta^{(t)} v^{(t)})^\top M_{i,j}(\tilde{w}^{(t)})(\eta^{(t)} v^{(t)}),$$

$$(36)$$

where $M_{i,j}(\tilde{w}^{(t)})$ is the Hessian matrices of $k_j(\cdot; i)$ at $\tilde{w}^{(t)}$ and $\tilde{w}^{(t)} = \alpha w^{(t)} + (1 - \alpha) w^{(t+1)}$ for some $\alpha \in [0, 1]$.

Shifting this back to the original function $h_j(\cdot; i)$ we have:

$$h_j(w^{(t+1)}; i) = k_j(w^{(t+1)}; i) + (h_j(w^{(t+1)}; i) - k_j(w^{(t+1)}; i))$$
$$\overset{(36)}{=} k_j(w^{(t)}; i) - \mathbf{J}_w k_j(w; i)|_{w=w^{(t)}} \eta^{(t)} v^{(t)} + \frac{1}{2} (\eta^{(t)} v^{(t)})^\top M_{i,j}(\tilde{w}^{(t)})(\eta^{(t)} v^{(t)})$$
$$+ (h_j(w^{(t+1)}; i) - k_j(w^{(t+1)}; i)),$$
$$= h_j(w^{(t)}; i) - \mathbf{J}_w k_j(w; i)|_{w=w^{(t)}} \eta^{(t)} v^{(t)} + \frac{1}{2} (\eta^{(t)} v^{(t)})^\top M_{i,j}(\tilde{w}^{(t)})(\eta^{(t)} v^{(t)})$$
$$+ (h_j(w^{(t+1)}; i) - k_j(w^{(t+1)}; i)) + (k_j(w^{(t)}; i) - h_j(w^{(t)}; i)),$$

which leads to our desired statement:

$$h(w^{(t+1)}; i) = h(w^{(t)} - \eta^{(t)} v^{(t)}; i) = h(w^{(t)}; i) - \eta^{(t)} H_i^{(t)} v^{(t)} + \epsilon_i^{(t)},$$

where

$$\epsilon_{i,j}^{(t)} = \frac{1}{2}(\eta^{(t)}v^{(t)})^\top M_{i,j}(\tilde{w}^{(t)})(\eta^{(t)}v^{(t)})$$
$$+ (h_j(w^{(t+1)};i) - k_j(w^{(t+1)};i)) + (k_j(w^{(t)};i) - h_j(w^{(t)};i)), \; j \in [c],$$

Hence we get the final bound:

$$|\epsilon_{i,j}^{(t)}| \leq \frac{1}{2}\left|(\eta^{(t)}v^{(t)})^\top M_{i,j}(\tilde{w}^{(t)})(\eta^{(t)}v^{(t)})\right|$$
$$+ |h_j(w^{(t+1)};i) - k_j(w^{(t+1)};i)| + |k_j(w^{(t)};i) - h_j(w^{(t)};i)|$$
$$\overset{(26)}{\leq} \frac{1}{2}\left|(\eta^{(t)}v^{(t)})^\top M_{i,j}(\tilde{w}^{(t)})(\eta^{(t)}v^{(t)})\right| + 2\varepsilon,$$
$$\leq \frac{1}{2}(\eta^{(t)})^2\|v^{(t)}\|^2 \cdot \|M_{i,j}(\tilde{w}^{(t)})\| + 2\varepsilon$$
$$\overset{(11)}{\leq} \frac{1}{2}(\eta^{(t)})^2\|v^{(t)}\|^2 G + 2\varepsilon, \; j \in [c].$$

$\square$

PROOF OF COROLLARY 1

*Proof.* The proof of this corollary follows directly by the applications of Lemmas 3 and 4.   $\square$

## E   TECHNICAL PROOFS FOR THEOREM 1

**Lemma 8.** *Suppose that Assumption 2 holds for $G > 0$ and Assumption 3 holds for $V > 0$, and $v^{(t)} = v^{(t)}_{* \, reg}$. Consider $\eta^{(t)} = D\sqrt{\varepsilon}$ for some $D > 0$ and $\varepsilon > 0$. For $i \in [n]$ and $0 \leq t < T$, we have*

$$\|\epsilon_i^{(t)}\|^2 \leq \frac{1}{4}c(4 + (V+2)GD^2)^2\varepsilon^2. \tag{37}$$

*Proof.* From (14), for $i \in [n]$, $j \in [c]$, and for $0 \leq t < T$, by Lemma 1 and Lemma 6 we have

$$|\epsilon_{i,j}^{(t)}| \leq \frac{1}{2}(\eta^{(t)})^2\|v^{(t)}\|^2 G + 2\varepsilon \leq \frac{1}{2}(V+2)GD^2\varepsilon + 2\varepsilon = \frac{1}{2}\varepsilon(4 + (V+2)GD^2),$$

where the last inequality follows by the fact $\|v^{(t)}\|^2 = \|v^{(t)}_{* \, reg}\|^2 \leq 2 + V$ of Lemma 2 and $\eta^{(t)} = D\sqrt{\varepsilon}$. Hence,

$$\|\epsilon_i^{(t)}\|^2 = \sum_{j=1}^{c}|\epsilon_{i,j}^{(t)}|^2 \leq \frac{1}{4}c(4 + (V+2)GD^2)^2\varepsilon^2.$$

$\square$

**Lemma 9.** *Let $w^{(t)}$ be generated by Algorithm 1 where we use the closed form solution for the search direction. We execute Algorithm 1 for $T = \frac{\beta}{\varepsilon}$ outer loops for some constant $\beta > 0$. We assume Assumption 1 holds. Suppose that Assumption 2 holds for $G > 0$ and Assumption 3 holds for $V > 0$. We set the step size equal to $\eta^{(t)} = D\sqrt{\varepsilon}$ for some $D > 0$ and choose a learning rate $\alpha_i^{(t)} \leq \frac{\alpha}{L_\phi}$, for some $\alpha \in (0, \frac{1}{3})$. For $i \in [n]$ and $0 \leq t < T$, we have*

$$\|h(w^{(t+1)};i) - h_i^*\|^2 \leq (1+\varepsilon)\|h(w^{(t)};i) - h_i^*\|^2 - 2(1-3\alpha)\alpha_i^{(t)}[\phi_i(h(w^{(t)};i)) - \phi_i(h_i^*)]$$
$$+ \frac{(3\varepsilon+2)}{4}c(4 + (V+2)GD^2)^2 \cdot \varepsilon$$
$$+ \frac{3\varepsilon+2}{\varepsilon}\|\eta^{(t)}H_i^{(t)}v^{(t)}_{* \, reg} - \alpha_i^{(t)}\nabla_z\phi_i(h(w^{(t)};i))\|^2 \tag{38}$$

*Proof.* Note that we have the optimal solution $v^{(t)}_{* \text{ reg}}$ for the optimization problem (15) for $0 \leq t < T$. From (12), we have, for $i \in [n]$,

$$
\begin{aligned}
h(w^{(t+1)}; i) &= h(w^{(t)} - \eta^{(t)} v^{(t)}_{* \text{ reg}}; i) \\
&= h(w^{(t)}; i) - \eta^{(t)} H^{(t)}_i v^{(t)}_{* \text{ reg}} + \epsilon^{(t)}_i \\
&= h(w^{(t)}; i) - \alpha^{(t)}_i \nabla_z \phi_i(h(w^{(t)}; i)) + \epsilon^{(t)}_i - [\eta^{(t)} H^{(t)}_i v^{(t)}_{* \text{ reg}} - \alpha^{(t)}_i \nabla_z \phi_i(h(w^{(t)}; i))].
\end{aligned}
$$

Hence, we have

$$
\begin{aligned}
&\|h(w^{(t+1)}; i) - h^*_i\|^2 \\
&= \|h(w^{(t)}; i) - h^*_i - \alpha^{(t)}_i \nabla_z \phi_i(h(w^{(t)}; i)) + \epsilon^{(t)}_i - [\eta^{(t)} H^{(t)}_i v^{(t)}_{* \text{ reg}} - \alpha^{(t)}_i \nabla_z \phi_i(h(w^{(t)}; i))]\|^2 \\
&= \|h(w^{(t)}; i) - h^*_i\|^2 + (\alpha^{(t)}_i)^2 \|\nabla_z \phi_i(h(w^{(t)}; i))\|^2 \\
&\quad + \|\epsilon^{(t)}_i\|^2 + \|\eta^{(t)} H^{(t)}_i v^{(t)}_{* \text{ reg}} - \alpha^{(t)}_i \nabla_z \phi_i(h(w^{(t)}; i))\|^2 \\
&\quad - 2 \cdot \langle h(w^{(t)}; i) - h^*_i, \alpha^{(t)}_i \nabla_z \phi_i(h(w^{(t)}; i)) \rangle \\
&\quad + 2 \cdot \langle h(w^{(t)}; i) - h^*_i, \epsilon^{(t)}_i \rangle \\
&\quad - 2 \cdot \langle h(w^{(t)}; i) - h^*_i, \eta^{(t)} H^{(t)}_i v^{(t)}_{* \text{ reg}} - \alpha^{(t)}_i \nabla_z \phi_i(h(w^{(t)}; i)) \rangle \\
&\quad - 2 \cdot \langle \alpha^{(t)}_i \nabla_z \phi_i(h(w^{(t)}; i)), \epsilon^{(t)}_i \rangle \\
&\quad + 2 \cdot \langle \alpha^{(t)}_i \nabla_z \phi_i(h(w^{(t)}; i)), \eta^{(t)} H^{(t)}_i v^{(t)}_{* \text{ reg}} - \alpha^{(t)}_i \nabla_z \phi_i(h(w^{(t)}; i)) \rangle \\
&\quad - 2 \cdot \langle \epsilon^{(t)}_i, \eta^{(t)} H^{(t)}_i v^{(t)}_{* \text{ reg}} - \alpha^{(t)}_i \nabla_z \phi_i(h(w^{(t)}; i)) \rangle,
\end{aligned}
$$

where we expand the square term. Now applying Young's inequalities: $2|\langle u, v \rangle| \leq \frac{\|u\|^2}{\varepsilon/2} + (\varepsilon/2)\|v\|^2$ for $\varepsilon > 0$ and $2|\langle u, v \rangle| \leq \|u\|^2 + \|v\|^2$ we have:

$$
\begin{aligned}
&\|h(w^{(t+1)}; i) - h^*_i\|^2 \\
&= \|h(w^{(t)}; i) - h^*_i\|^2 + (\alpha^{(t)}_i)^2 \|\nabla_z \phi_i(h(w^{(t)}; i))\|^2 \\
&\quad + \|\epsilon^{(t)}_i\|^2 + \|\eta^{(t)} H^{(t)}_i v^{(t)}_{* \text{ reg}} - \alpha^{(t)}_i \nabla_z \phi_i(h(w^{(t)}; i))\|^2 \\
&\quad - 2\alpha^{(t)}_i \langle h(w^{(t)}; i) - h^*_i, \nabla_z \phi_i(h(w^{(t)}; i)) \rangle \\
&\quad + \frac{\varepsilon}{2} \|h(w^{(t)}; i) - h^*_i\|^2 + \frac{2}{\varepsilon} \|\epsilon^{(t)}_i\|^2 \\
&\quad + \frac{\varepsilon}{2} \|h(w^{(t)}; i) - h^*_i\|^2 + \frac{2}{\varepsilon} \|\eta^{(t)} H^{(t)}_i v^{(t)}_{* \text{ reg}} - \alpha^{(t)}_i \nabla_z \phi_i(h(w^{(t)}; i))\|^2 \\
&\quad + 2(\alpha^{(t)}_i)^2 \|\nabla_z \phi_i(h(w^{(t)}; i))\|^2 + 2\|\epsilon^{(t)}_i\|^2 \\
&\quad + 2\|\eta^{(t)} H^{(t)}_i v^{(t)}_{* \text{ reg}} - \alpha^{(t)}_i \nabla_z \phi_i(h(w^{(t)}; i))\|^2 \\
&\overset{(22)}{\leq} (1+\varepsilon)\|h(w^{(t)}; i) - h^*_i\|^2 + 3(\alpha^{(t)}_i)^2 \|\nabla_z \phi_i(h(w^{(t)}; i))\|^2 \\
&\quad + \left(3 + \frac{2}{\varepsilon}\right) \|\epsilon^{(t)}_i\|^2 + \left(3 + \frac{2}{\varepsilon}\right) \|\eta^{(t)} H^{(t)}_i v^{(t)}_{* \text{ reg}} - \alpha^{(t)}_i \nabla_z \phi_i(h(w^{(t)}; i))\|^2 \\
&\quad - 2\alpha^{(t)}_i [\phi_i(h(w^{(t)}; i)) - \phi_i(h^*_i)]
\end{aligned}
$$

Note that from (23) we get that $\|\nabla_z \phi_i(h(w^{(t)}; i))\|^2 \leq 2L_\phi[\phi_i(h(w^{(t)}; i)) - \phi_i(h^*_i)]$. Applying this and using the fact that $\alpha^{(t)}_i \leq \frac{\alpha}{L_\phi}$, for some $\alpha \in (0, \frac{1}{3})$, we are able to derive:

$$
\begin{aligned}
&\|h(w^{(t+1)}; i) - h^*_i\|^2 \\
&\leq (1+\varepsilon)\|h(w^{(t)}; i) - h^*_i\|^2 - 2(1 - 3\alpha)\alpha^{(t)}_i [\phi_i(h(w^{(t)}; i)) - \phi_i(h^*_i)] \\
&\quad + \frac{3\varepsilon + 2}{\varepsilon} \|\epsilon^{(t)}_i\|^2 + \frac{3\varepsilon + 2}{\varepsilon} \|\eta^{(t)} H^{(t)}_i v^{(t)}_{* \text{ reg}} - \alpha^{(t)}_i \nabla_z \phi_i(h(w^{(t)}; i))\|^2 \\
&\leq (1+\varepsilon)\|h(w^{(t)}; i) - h^*_i\|^2 - 2(1 - 3\alpha)\alpha^{(t)}_i [\phi_i(h(w^{(t)}; i)) - \phi_i(h^*_i)]
\end{aligned}
$$

$$+ \frac{3\varepsilon + 2}{\varepsilon} \frac{1}{4} c(4 + (V+2)GD^2)^2 \varepsilon^2 + \frac{3\varepsilon + 2}{\varepsilon} \|\eta^{(t)} H_i^{(t)} v_{* \, \mathrm{reg}}^{(t)} - \alpha_i^{(t)} \nabla_z \phi_i(h(w^{(t)}; i))\|^2$$

where the last inequality follows by Lemma 8. $\qquad\square$

**Lemma 10.** *Let $w^{(t)}$ be generated by Algorithm 1 where we use the closed form solution for the search direction. We execute Algorithm 1 for $T = \frac{\beta}{\varepsilon}$ outer loops for some constant $\beta > 0$. We assume Assumption 1 holds. Suppose that Assumption 2 holds for $G > 0$ and Assumption 3 holds for $V > 0$. We set the step size equal to $\eta^{(t)} = D\sqrt{\varepsilon}$ for some $D > 0$ and choose a learning rate $\alpha_i^{(t)} = (1+\varepsilon)\alpha_i^{(t-1)} = (1+\varepsilon)^t \alpha_i^{(0)}$. Based on $\beta$, we define $\alpha_i^{(0)} = \frac{\alpha}{e^\beta L_\phi}$ with $\alpha \in (0, \frac{1}{3})$.*

*We have*

$$\frac{1}{T} \sum_{t=0}^{T-1} \frac{1}{n} \sum_{i=1}^{n} [f(w^{(t)}; i) - \phi_i(h_i^*)] \leq \frac{e^\beta L_\phi (1+\varepsilon)}{2(1-3\alpha)\alpha\beta} \cdot \frac{1}{n} \sum_{i=1}^{n} \|h(w^{(0)}; i) - h_i^*\|^2 \cdot \varepsilon$$

$$+ \frac{e^\beta L_\phi}{8\alpha(1-3\alpha)} (3\varepsilon + 2) \left[ c(4 + (V+2)GD^2)^2 + 8 + 4V \right] \cdot \varepsilon. \tag{39}$$

*Proof.* Rearranging the terms in Lemma 9, we have

$$\phi_i(h(w^{(t)}; i)) - \phi_i(h_i^*) \leq \frac{1}{2(1-3\alpha)} \left( \frac{(1+\varepsilon)}{\alpha_i^{(t)}} \|h(w^{(t)}; i) - h_i^*\|^2 - \frac{1}{\alpha_i^{(t)}} \|h(w^{(t+1)}; i) - h_i^*\|^2 \right)$$

$$+ \frac{1}{8(1-3\alpha)} \cdot \frac{1}{\alpha_i^{(t)}} \cdot \varepsilon(3\varepsilon + 2)c(4 + (V+2)GD^2)^2$$

$$+ \frac{1}{2(1-3\alpha)} \cdot \frac{1}{\alpha_i^{(t)}} \cdot \frac{3\varepsilon + 2}{\varepsilon} \|\eta^{(t)} H_i^{(t)} v_{* \, \mathrm{reg}}^{(t)} - \alpha_i^{(t)} \nabla_z \phi_i(h(w^{(t)}; i))\|^2$$

$$\leq \frac{1}{2(1-3\alpha)} \left( \frac{(1+\varepsilon)}{\alpha_i^{(t)}} \|h(w^{(t)}; i) - h_i^*\|^2 - \frac{(1+\varepsilon)}{\alpha_i^{(t+1)}} \|h(w^{(t+1)}; i) - h_i^*\|^2 \right)$$

$$+ \frac{e^\beta L_\phi}{8\alpha(1-3\alpha)} \cdot \varepsilon(3\varepsilon + 2)c(4 + (V+2)GD^2)^2$$

$$+ \frac{e^\beta L_\phi}{2\alpha(1-3\alpha)} \cdot \frac{3\varepsilon + 2}{\varepsilon} \|\eta^{(t)} H_i^{(t)} v_{* \, \mathrm{reg}}^{(t)} - \alpha_i^{(t)} \nabla_z \phi_i(h(w^{(t)}; i))\|^2.$$

The last inequality follows because the learning rate satisfies $\alpha_i^{(0)} = \frac{\alpha}{e^\beta L_\phi} \leq \frac{\alpha}{L_\phi}$ and for $t = 1, \ldots, T = \frac{\beta}{\varepsilon}$ for some $\beta > 0$

$$\alpha_i^{(t)} = (1+\varepsilon)\alpha_i^{(t-1)} = (1+\varepsilon)^t \alpha_i^{(0)} \leq (1+\varepsilon)^T \alpha_i^{(0)} = (1+\varepsilon)^{\beta/\varepsilon} \frac{\alpha}{e^\beta L_\phi} \leq \frac{\alpha}{L_\phi},$$

since $(1+x)^{1/x} \leq e$, $x > 0$. Moreover, we have $\frac{1}{\alpha_i^{(t)}} \leq \frac{1}{\alpha_i^{(0)}} = \frac{e^\beta L_\phi}{\alpha}$, $t = 0, \ldots, T-1$.

Taking the average sum from $t = 0, \ldots, T-1$, we have

$$\frac{1}{T} \sum_{t=0}^{T-1} [\phi_i(h(w^{(t)}; i)) - \phi_i(h_i^*)] \leq \frac{1}{2(1-3\alpha)T} \cdot \frac{(1+\varepsilon)}{\alpha_i^{(0)}} \|h(w^{(0)}; i) - h_i^*\|^2$$

$$+ \frac{e^\beta L_\phi}{8\alpha(1-3\alpha)} \cdot \varepsilon(3\varepsilon + 2)c(4 + (V+2)GD^2)^2$$

$$+ \frac{e^\beta L_\phi}{2\alpha(1-3\alpha)} \cdot \frac{3\varepsilon + 2}{\varepsilon} \frac{1}{T} \sum_{t=0}^{T-1} \|\eta^{(t)} H_i^{(t)} v_{* \, \mathrm{reg}}^{(t)} - \alpha_i^{(t)} \nabla_z \phi_i(h(w^{(t)}; i))\|^2$$

$$= \frac{e^\beta L_\phi (1+\varepsilon)}{2(1-3\alpha)\alpha\beta} \varepsilon \cdot \|h(w^{(0)}; i) - h_i^*\|^2$$

$$+ \frac{e^\beta L_\phi}{8\alpha(1-3\alpha)} \cdot \varepsilon(3\varepsilon+2)c(4+(V+2)GD^2)^2$$

$$+ \frac{e^\beta L_\phi}{2\alpha(1-3\alpha)} \cdot \frac{3\varepsilon+2}{\varepsilon} \frac{1}{T} \sum_{t=0}^{T-1} \|\eta^{(t)} H_i^{(t)} v_{*\,\mathrm{reg}}^{(t)} - \alpha_i^{(t)} \nabla_z \phi_i(h(w^{(t)};i))\|^2.$$

Taking the average sum from $i = 1, \ldots, n$, we have

$$\frac{1}{T} \sum_{t=0}^{T-1} \frac{1}{n} \sum_{i=1}^{n} [\phi_i(h(w^{(t)};i)) - \phi_i(h_i^*)]$$

$$\leq \frac{e^\beta L_\phi(1+\varepsilon)}{2(1-3\alpha)\alpha\beta} \varepsilon \cdot \frac{1}{n} \sum_{i=1}^{n} \|h(w^{(0)};i) - h_i^*\|^2$$

$$+ \frac{e^\beta L_\phi}{8\alpha(1-3\alpha)} \cdot \varepsilon(3\varepsilon+2)c(4+(V+2)GD^2)^2$$

$$+ \frac{e^\beta L_\phi}{2\alpha(1-3\alpha)} \cdot \frac{3\varepsilon+2}{\varepsilon} \frac{1}{T} \sum_{t=0}^{T-1} \frac{1}{n} \sum_{i=1}^{n} \|\eta^{(t)} H_i^{(t)} v_{*\,\mathrm{reg}}^{(t)} - \alpha_i^{(t)} \nabla_z \phi_i(h(w^{(t)};i))\|^2$$

$$\overset{(17)}{\leq} \frac{e^\beta L_\phi(1+\varepsilon)}{2(1-3\alpha)\alpha\beta} \varepsilon \cdot \frac{1}{n} \sum_{i=1}^{n} \|h(w^{(0)};i) - h_i^*\|^2$$

$$+ \frac{e^\beta L_\phi}{8\alpha(1-3\alpha)} \cdot \varepsilon(3\varepsilon+2)c(4+(V+2)GD^2)^2$$

$$+ \frac{e^\beta L_\phi}{2\alpha(1-3\alpha)} \cdot \frac{3\varepsilon+2}{\varepsilon}(2+V)\varepsilon^2. \tag{40}$$

Note that

$$\frac{1}{T} \sum_{t=0}^{T-1} \frac{1}{n} \sum_{i=1}^{n} [\phi_i(h(w^{(t)};i)) - \phi_i(h_i^*)] = \frac{1}{T} \sum_{t=0}^{T-1} \frac{1}{n} \sum_{i=1}^{n} [f(w^{(t)};i) - \phi_i(h_i^*)]. \tag{41}$$

Therefore, applying (41) to (40), we have

$$\frac{1}{T} \sum_{t=0}^{T-1} \frac{1}{n} \sum_{i=1}^{n} [f(w^{(t)};i) - \phi_i(h_i^*)]$$

$$\leq \frac{e^\beta L_\phi(1+\varepsilon)}{2(1-3\alpha)\alpha\beta} \cdot \frac{1}{n} \sum_{i=1}^{n} \|h(w^{(0)};i) - h_i^*\|^2 \cdot \varepsilon$$

$$+ \frac{e^\beta L_\phi}{8\alpha(1-3\alpha)}(3\varepsilon+2)\left[c(4+(V+2)GD^2)^2 + 8 + 4V\right] \cdot \varepsilon.$$

which is our desired result. $\qquad\square$

PROOF OF THEOREM 1

*Proof.* We have

$$F_* = \min_{w \in \mathbb{R}^d} F(w) = \min_{w \in \mathbb{R}^d} \left( \frac{1}{n} \sum_{i=1}^{n} f_i(w) \right) = \frac{1}{n} \min_{w \in \mathbb{R}^d} \left( \sum_{i=1}^{n} f_i(w) \right)$$

$$\geq \frac{1}{n} \sum_{i=1}^{n} \min_{w \in \mathbb{R}^d} (f_i(w)) = \frac{1}{n} \sum_{i=1}^{n} f_i^* \geq \frac{1}{n} \sum_{i=1}^{n} \phi_i(h_i^*). \tag{42}$$

Hence $F_* - \frac{1}{n} \sum_{i=1}^{n} \phi_i(h_i^*) \geq 0$. Therefore

$$\frac{1}{T} \sum_{t=1}^{T} [F(w^{(t)}) - F_*] = \frac{1}{T} \sum_{t=1}^{T} \left( \frac{1}{n} \sum_{i=1}^{n} [f(w^{(t)};i) - \phi_i(h_i^*)] - \left[ F_* - \frac{1}{n} \sum_{i=1}^{n} \phi_i(h_i^*) \right] \right)$$

$$\leq \frac{1}{T} \sum_{t=1}^{T} \frac{1}{n} \sum_{i=1}^{n} [f(w^{(t)}; i) - \phi_i(h_i^*)]$$

$$\overset{(39)}{\leq} \frac{e^\beta L_\phi (1+\varepsilon)}{2(1-3\alpha)\alpha\beta} \cdot \frac{1}{n} \sum_{i=1}^{n} \|h(w^{(0)}; i) - h_i^*\|^2 \cdot \varepsilon$$

$$+ \frac{e^\beta L_\phi (3\varepsilon + 2)}{8\alpha(1-3\alpha)} \left[ c(4 + (V+2)GD^2)^2 + 8 + 4V \right] \cdot \varepsilon.$$

□

## F    TECHNICAL PROOFS FOR THEOREM 2

**Lemma 11.** *For $0 \leq t < T$, suppose that Assumption 3 holds for $V \geq 0$ and $v^{(t)}$ satisfies (19). Then*

$$\|v^{(t)}\|^2 \leq 2(\varepsilon^2 + V + 2).$$

*Proof.* From $\|v^{(t)} - v_{* \text{ reg}}^{(t)}\| \leq \varepsilon$. Using $\|a\|^2 \leq 2\|a - b\|^2 + 2\|b\|^2$, we have

$$\|v^{(t)}\|^2 \leq 2\|v^{(t)} - v_{* \text{ reg}}^{(t)}\|^2 + 2\|v_{* \text{ reg}}^{(t)}\|^2 \overset{(19)}{\leq} 2\varepsilon^2 + 4 + 2V.$$

where the last inequality follows since $\|v_{* \text{ reg}}^{(t)}\|^2 \leq 2 + V$ for some $V > 0$ in Lemma 2.

□

**Lemma 12.** *Suppose that Assumption 2 holds for $G > 0$ and Assumption 3 holds for $V > 0$. Consider $\eta^{(t)} = D\sqrt{\varepsilon}$ for some $D > 0$ and $\varepsilon > 0$. For $i \in [n]$ and $0 \leq t < T$, we have*

$$\|\epsilon_i^{(t)}\|^2 \leq c(2 + (V + \varepsilon^2 + 2)GD^2)^2 \varepsilon^2. \tag{43}$$

*Proof.* From (14), for $i \in [n]$, $j \in [c]$, and for $0 \leq t < T$, by Lemma 1 and Lemma 6 we have

$$|\epsilon_{i,j}^{(t)}| \leq \frac{1}{2}(\eta^{(t)})^2 \|v^{(t)}\|^2 G + 2\varepsilon \leq \frac{1}{2} 2(\varepsilon^2 + V + 2)GD^2 \varepsilon + 2\varepsilon = \varepsilon(2 + (V + \varepsilon^2 + 2)GD^2),$$

where the last inequality follows by the application of Lemma 11 and $\eta^{(t)} = D\sqrt{\varepsilon}$. Hence,

$$\|\epsilon_i^{(t)}\|^2 = \sum_{j=1}^{c} |\epsilon_{i,j}^{(t)}|^2 \leq c(2 + (V + \varepsilon^2 + 2)GD^2)^2 \varepsilon^2.$$

□

**Lemma 13.** *Let $w^{(t)}$ be generated by Algorithm 2 where $v^{(t)}$ satisfies (19). We execute Algorithm 2 for $T = \frac{\beta}{\varepsilon}$ outer loops for some constant $\beta > 0$. We assume Assumption 1 holds. Suppose that Assumption 2 holds for $G > 0$, Assumption 3 holds for $V > 0$ and Assumption 4 holds for $H > 0$. We set the step size equal to $\eta^{(t)} = D\sqrt{\varepsilon}$ for some $D > 0$ and choose a learning rate $\alpha_i^{(t)} \leq \frac{\alpha}{L_\phi}$, for some $\alpha \in (0, \frac{1}{4})$. For $i \in [n]$ and $0 \leq t < T$, we have*

$$\|h(w^{(t+1)}; i) - h_i^*\|^2 \leq (1+\varepsilon)\|h(w^{(t)}; i) - h_i^*\|^2 - 2(1-4\alpha)\alpha_i^{(t)}[\phi_i(h(w^{(t)}; i)) - \phi_i(h_i^*)]$$

$$+ \varepsilon(4\varepsilon + 3) \left[ D^2 H^2 + c(2 + (V + \varepsilon^2 + 2)GD^2)^2 \right]$$

$$+ \frac{4\varepsilon + 3}{\varepsilon} \|\eta^{(t)} H_i^{(t)} v_{* \text{ reg}}^{(t)} - \alpha_i^{(t)} \nabla_z \phi_i(h(w^{(t)}; i))\|^2 \tag{44}$$

*Proof.* Note that $v^{(t)}$ is obtained from the optimization problem (15) for $0 \leq t < T$. From (9), we have, for $i \in [n]$,

$$
\begin{aligned}
h(w^{(t+1)}; i) &= h(w^{(t)} - \eta^{(t)} v^{(t)}; i) \\
&= h(w^{(t)}; i) - \eta^{(t)} H_i^{(t)} v^{(t)} + \epsilon_i^{(t)} \\
&= h(w^{(t)}; i) - \eta^{(t)} H_i^{(t)} (v^{(t)} - v_{* \text{ reg}}^{(t)}) - \alpha_i^{(t)} \nabla_z \phi_i(h(w^{(t)}; i)) + \epsilon_i^{(t)} \\
&\quad - [\eta^{(t)} H_i^{(t)} v_{* \text{ reg}}^{(t)} - \alpha_i^{(t)} \nabla_z \phi_i(h(w^{(t)}; i))].
\end{aligned}
$$

Hence, we have

$$
\begin{aligned}
&\| h(w^{(t+1)}; i) - h_i^* \|^2 \\
&= \| h(w^{(t)}; i) - h_i^* - \eta^{(t)} H_i^{(t)} (v^{(t)} - v_{* \text{ reg}}^{(t)}) - \alpha_i^{(t)} \nabla_z \phi_i(h(w^{(t)}; i)) \\
&\quad + \epsilon_i^{(t)} - [\eta^{(t)} H_i^{(t)} v_{* \text{ reg}}^{(t)} - \alpha_i^{(t)} \nabla_z \phi_i(h(w^{(t)}; i))] \|^2 \\
&= \| h(w^{(t)}; i) - h_i^* \|^2 + \| \eta^{(t)} H_i^{(t)} (v^{(t)} - v_{* \text{ reg}}^{(t)}) \|^2 + (\alpha_i^{(t)})^2 \| \nabla_z \phi_i(h(w^{(t)}; i)) \|^2 \\
&\quad + \| \epsilon_i^{(t)} \|^2 + \| \eta^{(t)} H_i^{(t)} v_{* \text{ reg}}^{(t)} - \alpha_i^{(t)} \nabla_z \phi_i(h(w^{(t)}; i)) \|^2 \\
&\quad - 2 \cdot \langle h(w^{(t)}; i) - h_i^*, \eta^{(t)} H_i^{(t)} (v^{(t)} - v_{* \text{ reg}}^{(t)}) \rangle \\
&\quad - 2 \cdot \langle h(w^{(t)}; i) - h_i^*, \alpha_i^{(t)} \nabla_z \phi_i(h(w^{(t)}; i)) \rangle \\
&\quad + 2 \cdot \langle h(w^{(t)}; i) - h_i^*, \epsilon_i^{(t)} \rangle \\
&\quad - 2 \cdot \langle h(w^{(t)}; i) - h_i^*, \eta^{(t)} H_i^{(t)} v_{* \text{ reg}}^{(t)} - \alpha_i^{(t)} \nabla_z \phi_i(h(w^{(t)}; i)) \rangle \\
&\quad + 2 \cdot \langle \eta^{(t)} H_i^{(t)} (v^{(t)} - v_{* \text{ reg}}^{(t)}), \alpha_i^{(t)} \nabla_z \phi_i(h(w^{(t)}; i)) \rangle \\
&\quad - 2 \cdot \langle \eta^{(t)} H_i^{(t)} (v^{(t)} - v_{* \text{ reg}}^{(t)}), \epsilon_i^{(t)} \rangle \\
&\quad + 2 \cdot \langle \eta^{(t)} H_i^{(t)} (v^{(t)} - v_{* \text{ reg}}^{(t)}), \eta^{(t)} H_i^{(t)} v_{* \text{ reg}}^{(t)} - \alpha_i^{(t)} \nabla_z \phi_i(h(w^{(t)}; i)) \rangle \\
&\quad - 2 \cdot \langle \alpha_i^{(t)} \nabla_z \phi_i(h(w^{(t)}; i)), \epsilon_i^{(t)} \rangle \\
&\quad + 2 \cdot \langle \alpha_i^{(t)} \nabla_z \phi_i(h(w^{(t)}; i)), \eta^{(t)} H_i^{(t)} v_{* \text{ reg}}^{(t)} - \alpha_i^{(t)} \nabla_z \phi_i(h(w^{(t)}; i)) \rangle \\
&\quad - 2 \cdot \langle \epsilon_i^{(t)}, \eta^{(t)} H_i^{(t)} v_{* \text{ reg}}^{(t)} - \alpha_i^{(t)} \nabla_z \phi_i(h(w^{(t)}; i)) \rangle,
\end{aligned}
$$

where we expand the square term. Now applying Young's inequalities: $2|\langle u, v \rangle| \leq \frac{\|u\|^2}{\varepsilon/3} + (\varepsilon/3)\|v\|^2$ for $\varepsilon > 0$ and $2|\langle u, v \rangle| \leq \|u\|^2 + \|v\|^2$ we have:

$$
\begin{aligned}
&\| h(w^{(t+1)}; i) - h_i^* \|^2 \\
&= \| h(w^{(t)}; i) - h_i^* \|^2 + \| \eta^{(t)} H_i^{(t)} (v^{(t)} - v_{* \text{ reg}}^{(t)}) \|^2 + (\alpha_i^{(t)})^2 \| \nabla_z \phi_i(h(w^{(t)}; i)) \|^2 \\
&\quad + \| \epsilon_i^{(t)} \|^2 + \| \eta^{(t)} H_i^{(t)} v_{* \text{ reg}}^{(t)} - \alpha_i^{(t)} \nabla_z \phi_i(h(w^{(t)}; i)) \|^2 \\
&\quad + \frac{\varepsilon}{3} \| h(w^{(t)}; i) - h_i^* \|^2 + \frac{3}{\varepsilon} \| \eta^{(t)} H_i^{(t)} (v^{(t)} - v_{* \text{ reg}}^{(t)}) \|^2 \\
&\quad - 2\alpha_i^{(t)} \langle h(w^{(t)}; i) - h_i^*, \nabla_z \phi_i(h(w^{(t)}; i)) \rangle \\
&\quad + \frac{\varepsilon}{3} \| h(w^{(t)}; i) - h_i^* \|^2 + \frac{3}{\varepsilon} \| \epsilon_i^{(t)} \|^2 \\
&\quad + \frac{\varepsilon}{3} \| h(w^{(t)}; i) - h_i^* \|^2 + \frac{3}{\varepsilon} \| \eta^{(t)} H_i^{(t)} v_{* \text{ reg}}^{(t)} - \alpha_i^{(t)} \nabla_z \phi_i(h(w^{(t)}; i)) \|^2 \\
&\quad + 3(\eta^{(t)})^2 \| H_i^{(t)} (v^{(t)} - v_{* \text{ reg}}^{(t)}) \|^2 + 3(\alpha_i^{(t)})^2 \| \nabla_z \phi_i(h(w^{(t)}; i)) \|^2 + 3 \| \epsilon_i^{(t)} \|^2 \\
&\quad + 3 \| \eta^{(t)} H_i^{(t)} v_{* \text{ reg}}^{(t)} - \alpha_i^{(t)} \nabla_z \phi_i(h(w^{(t)}; i)) \|^2 \\
&\overset{(22)}{\leq} (1 + \varepsilon) \| h(w^{(t)}; i) - h_i^* \|^2 + 4(\alpha_i^{(t)})^2 \| \nabla_z \phi_i(h(w^{(t)}; i)) \|^2 \\
&\quad + \left( 4 + \frac{3}{\varepsilon} \right) \| \eta^{(t)} H_i^{(t)} (v^{(t)} - v_{* \text{ reg}}^{(t)}) \|^2 + \left( 4 + \frac{3}{\varepsilon} \right) \| \epsilon_i^{(t)} \|^2
\end{aligned}
$$

$$+ \left(4 + \frac{3}{\varepsilon}\right) \|\eta^{(t)} H_i^{(t)} v_{* \text{ reg}}^{(t)} - \alpha_i^{(t)} \nabla_z \phi_i(h(w^{(t)}; i))\|^2$$

$$- 2\alpha_i^{(t)} [\phi_i(h(w^{(t)}; i)) - \phi_i(h_i^*)]$$

Note that from (23) we get that $\|\nabla_z \phi_i(h(w^{(t)}; i))\|^2 \leq 2L_\phi [\phi_i(h(w^{(t)}; i)) - \phi_i(h_i^*)]$. Applying this and using the fact that $\alpha_i^{(t)} \leq \frac{\alpha}{L_\phi}$, for some $\alpha \in (0, \frac{1}{4})$, we are able to derive:

$$\|h(w^{(t+1)}; i) - h_i^*\|^2$$

$$\leq (1 + \varepsilon)\|h(w^{(t)}; i) - h_i^*\|^2 - 2(1 - 4\alpha)\alpha_i^{(t)}[\phi_i(h(w^{(t)}; i)) - \phi_i(h_i^*)]$$

$$+ \frac{4\varepsilon + 3}{\varepsilon}\|\eta^{(t)} H_i^{(t)}(v^{(t)} - v_{* \text{ reg}}^{(t)})\|^2 + \frac{4\varepsilon + 3}{\varepsilon}\|\epsilon_i^{(t)}\|^2$$

$$+ \frac{4\varepsilon + 3}{\varepsilon}\|\eta^{(t)} H_i^{(t)} v_{* \text{ reg}}^{(t)} - \alpha_i^{(t)} \nabla_z \phi_i(h(w^{(t)}; i))\|^2$$

$$\overset{(a)}{\leq} (1 + \varepsilon)\|h(w^{(t)}; i) - h_i^*\|^2 - 2(1 - 4\alpha)\alpha_i^{(t)}[\phi_i(h(w^{(t)}; i)) - \phi_i(h_i^*)]$$

$$+ \frac{4\varepsilon + 3}{\varepsilon} D^2 \varepsilon \frac{H^2}{\varepsilon}\|v^{(t)} - v_{* \text{ reg}}^{(t)}\|^2 + \frac{4\varepsilon + 3}{\varepsilon}\|\epsilon_i^{(t)}\|^2$$

$$+ \frac{4\varepsilon + 3}{\varepsilon}\|\eta^{(t)} H_i^{(t)} v_{* \text{ reg}}^{(t)} - \alpha_i^{(t)} \nabla_z \phi_i(h(w^{(t)}; i))\|^2$$

$$\overset{(b)}{\leq} (1 + \varepsilon)\|h(w^{(t)}; i) - h_i^*\|^2 - 2(1 - 4\alpha)\alpha_i^{(t)}[\phi_i(h(w^{(t)}; i)) - \phi_i(h_i^*)]$$

$$+ \frac{4\varepsilon + 3}{\varepsilon} D^2 H^2 \cdot \varepsilon^2 + \frac{4\varepsilon + 3}{\varepsilon} \cdot c(2 + (V + \varepsilon^2 + 2)GD^2)^2 \varepsilon^2$$

$$+ \frac{4\varepsilon + 3}{\varepsilon}\|\eta^{(t)} H_i^{(t)} v_{* \text{ reg}}^{(t)} - \alpha_i^{(t)} \nabla_z \phi_i(h(w^{(t)}; i))\|^2$$

$$= (1 + \varepsilon)\|h(w^{(t)}; i) - h_i^*\|^2 - 2(1 - 4\alpha)\alpha_i^{(t)}[\phi_i(h(w^{(t)}; i)) - \phi_i(h_i^*)]$$

$$+ \varepsilon(4\varepsilon + 3)\left[D^2 H^2 + c(2 + (V + \varepsilon^2 + 2)GD^2)^2\right]$$

$$+ \frac{4\varepsilon + 3}{\varepsilon}\|\eta^{(t)} H_i^{(t)} v_{* \text{ reg}}^{(t)} - \alpha_i^{(t)} \nabla_z \phi_i(h(w^{(t)}; i))\|^2$$

where $(a)$ follows by using matrix vector inequality $\|Hv\| \leq \|H\|\|v\|$, where $H \in \mathbb{R}^{c \times d}$ and $v \in \mathbb{R}^d$ and Assumption 4 in (20) and $\eta^{(t)} = D\sqrt{\varepsilon}$ for some $D > 0$ and $\varepsilon > 0$; $(b)$ follows by the fact that $\|v^{(t)} - v_{* \text{ reg}}^{(t)}\|^2 \leq \varepsilon^2$ in (19) and Lemma 12. $\qquad \square$

**Lemma 14.** *Let $w^{(t)}$ be generated by Algorithm 2 where $v^{(t)}$ satisfies (19). We execute Algorithm 2 for $T = \frac{\beta}{\varepsilon}$ outer loops for some constant $\beta > 0$. We assume Assumption 1 holds. Suppose that Assumption 2 holds for $G > 0$, Assumption 3 holds for $V > 0$ and Assumption 4 holds for $H > 0$. We set the step size equal to $\eta^{(t)} = D\sqrt{\varepsilon}$ for some $D > 0$ and choose a learning rate $\alpha_i^{(t)} = (1 + \varepsilon)\alpha_i^{(t-1)} = (1 + \varepsilon)^t \alpha_i^{(0)}$. Based on $\beta$, we define $\alpha_i^{(0)} = \frac{\alpha}{e^\beta L_\phi}$ with $\alpha \in (0, \frac{1}{4})$.*

*We have*

$$\frac{1}{T}\sum_{t=0}^{T-1}\frac{1}{n}\sum_{i=1}^{n}[f(w^{(t)}; i) - \phi_i(h_i^*)]$$

$$\leq \frac{e^\beta L_\phi (1 + \varepsilon)}{2(1 - 4\alpha)\alpha\beta} \cdot \frac{1}{n}\sum_{i=1}^{n}\|h(w^{(0)}; i) - h_i^*\|^2 \cdot \varepsilon$$

$$+ \frac{e^\beta L_\phi (4\varepsilon + 3)}{2\alpha(1 - 4\alpha)}\left[D^2 H^2 + c(2 + (V + \varepsilon^2 + 2)GD^2)^2 + 2 + V\right] \cdot \varepsilon. \qquad (45)$$

*Proof.* Rearranging the terms in Lemma 13, we have

$$\phi_i(h(w^{(t)}; i)) - \phi_i(h_i^*) \leq \frac{1}{2(1 - 4\alpha)}\left(\frac{(1 + \varepsilon)}{\alpha_i^{(t)}}\|h(w^{(t)}; i) - h_i^*\|^2 - \frac{1}{\alpha_i^{(t)}}\|h(w^{(t+1)}; i) - h_i^*\|^2\right)$$

$$+ \frac{1}{2(1-4\alpha)} \cdot \frac{1}{\alpha_i^{(t)}} \cdot \varepsilon(4\varepsilon+3) \left[ D^2 H^2 + c(2 + (V + \varepsilon^2 + 2)GD^2)^2 \right]$$

$$+ \frac{1}{2(1-4\alpha)} \cdot \frac{1}{\alpha_i^{(t)}} \cdot \frac{4\varepsilon+3}{\varepsilon} \| \eta^{(t)} H_i^{(t)} v_{* \, \mathrm{reg}}^{(t)} - \alpha_i^{(t)} \nabla_z \phi_i(h(w^{(t)};i)) \|^2$$

$$\leq \frac{1}{2(1-4\alpha)} \left( \frac{(1+\varepsilon)}{\alpha_i^{(t)}} \| h(w^{(t)};i) - h_i^* \|^2 - \frac{(1+\varepsilon)}{\alpha_i^{(t+1)}} \| h(w^{(t+1)};i) - h_i^* \|^2 \right)$$

$$+ \frac{e^\beta L_\phi}{2\alpha(1-4\alpha)} \cdot \varepsilon(4\varepsilon+3) \left[ D^2 H^2 + c(2 + (V + \varepsilon^2 + 2)GD^2)^2 \right]$$

$$+ \frac{e^\beta L_\phi}{2\alpha(1-4\alpha)} \cdot \frac{4\varepsilon+3}{\varepsilon} \| \eta^{(t)} H_i^{(t)} v_{* \, \mathrm{reg}}^{(t)} - \alpha_i^{(t)} \nabla_z \phi_i(h(w^{(t)};i)) \|^2.$$

$$(46)$$

The last inequality follows because the learning rate satisfies $\alpha_i^{(0)} = \frac{\alpha}{e^\beta L_\phi} \leq \frac{\alpha}{L_\phi}$ and for $t = 1, \ldots, T = \frac{\beta}{\varepsilon}$ for some $\beta > 0$

$$\alpha_i^{(t)} = (1+\varepsilon)\alpha_i^{(t-1)} = (1+\varepsilon)^t \alpha_i^{(0)} \leq (1+\varepsilon)^T \alpha_i^{(0)} = (1+\varepsilon)^{\beta/\varepsilon} \frac{\alpha}{e^\beta L_\phi} \leq \frac{\alpha}{L_\phi},$$

since $(1+x)^{1/x} \leq e, x > 0$. Moreover, we have $\frac{1}{\alpha_i^{(t)}} \leq \frac{1}{\alpha_i^{(0)}} = \frac{e^\beta L_\phi}{\alpha}, t = 0, \ldots, T-1$.

Taking the average sum from $t = 0, \ldots, T-1$, we have

$$\frac{1}{T} \sum_{t=0}^{T-1} [\phi_i(h(w^{(t)};i)) - \phi_i(h_i^*)] \leq \frac{1}{2(1-4\alpha)T} \cdot \frac{(1+\varepsilon)}{\alpha_i^{(0)}} \| h(w^{(0)};i) - h_i^* \|^2$$

$$+ \frac{e^\beta L_\phi}{2\alpha(1-4\alpha)} \cdot \varepsilon(4\varepsilon+3) \left[ D^2 H^2 + c(2 + (V + \varepsilon^2 + 2)GD^2)^2 \right]$$

$$+ \frac{e^\beta L_\phi}{2\alpha(1-4\alpha)} \cdot \frac{4\varepsilon+3}{\varepsilon} \frac{1}{T} \sum_{t=0}^{T-1} \| \eta^{(t)} H_i^{(t)} v_{* \, \mathrm{reg}}^{(t)} - \alpha_i^{(t)} \nabla_z \phi_i(h(w^{(t)};i)) \|^2$$

$$= \frac{e^\beta L_\phi (1+\varepsilon)}{2(1-4\alpha)\alpha\beta} \varepsilon \cdot \| h(w^{(0)};i) - h_i^* \|^2$$

$$+ \frac{e^\beta L_\phi}{2\alpha(1-4\alpha)} \cdot \varepsilon(4\varepsilon+3) \left[ D^2 H^2 + c(2 + (V + \varepsilon^2 + 2)GD^2)^2 \right]$$

$$+ \frac{e^\beta L_\phi}{2\alpha(1-4\alpha)} \cdot \frac{4\varepsilon+3}{\varepsilon} \frac{1}{T} \sum_{t=0}^{T-1} \| \eta^{(t)} H_i^{(t)} v_{* \, \mathrm{reg}}^{(t)} - \alpha_i^{(t)} \nabla_z \phi_i(h(w^{(t)};i)) \|^2.$$

Taking the average sum from $i = 1, \ldots, n$, we have

$$\frac{1}{T} \sum_{t=0}^{T-1} \frac{1}{n} \sum_{i=1}^{n} [\phi_i(h(w^{(t)};i)) - \phi_i(h_i^*)]$$

$$\leq \frac{e^\beta L_\phi (1+\varepsilon)}{2(1-4\alpha)\alpha\beta} \varepsilon \cdot \frac{1}{n} \sum_{i=1}^{n} \| h(w^{(0)};i) - h_i^* \|^2$$

$$+ \frac{e^\beta L_\phi}{2\alpha(1-4\alpha)} \cdot \varepsilon(4\varepsilon+3) \left[ D^2 H^2 + c(2 + (V + \varepsilon^2 + 2)GD^2)^2 \right]$$

$$+ \frac{e^\beta L_\phi}{2\alpha(1-4\alpha)} \cdot \frac{4\varepsilon+3}{\varepsilon} \frac{1}{T} \sum_{t=0}^{T-1} \frac{1}{n} \sum_{i=1}^{n} \| \eta^{(t)} H_i^{(t)} v_{* \, \mathrm{reg}}^{(t)} - \alpha_i^{(t)} \nabla_z \phi_i(h(w^{(t)};i)) \|^2$$

$$\overset{(17)}{\leq} \frac{e^\beta L_\phi (1+\varepsilon)}{2(1-4\alpha)\alpha\beta} \varepsilon \cdot \frac{1}{n} \sum_{i=1}^{n} \| h(w^{(0)};i) - h_i^* \|^2$$

$$+ \frac{e^\beta L_\phi}{2\alpha(1-4\alpha)} \cdot \varepsilon(4\varepsilon + 3)\left[D^2 H^2 + c(2 + (V + \varepsilon^2 + 2)GD^2)^2\right]$$
$$+ \frac{e^\beta L_\phi}{2\alpha(1-4\alpha)} \cdot \frac{4\varepsilon + 3}{\varepsilon}(2 + V)\varepsilon^2. \tag{47}$$

Note that

$$\frac{1}{T}\sum_{t=0}^{T-1}\frac{1}{n}\sum_{i=1}^{n}[\phi_i(h(w^{(t)};i)) - \phi_i(h_i^*)] = \frac{1}{T}\sum_{t=0}^{T-1}\frac{1}{n}\sum_{i=1}^{n}[f(w^{(t)};i) - \phi_i(h_i^*)]. \tag{48}$$

Therefore, applying (48) to (47), we have

$$\frac{1}{T}\sum_{t=0}^{T-1}\frac{1}{n}\sum_{i=1}^{n}[f(w^{(t)};i) - \phi_i(h_i^*)]$$
$$\leq \frac{e^\beta L_\phi(1+\varepsilon)}{2(1-4\alpha)\alpha\beta} \cdot \frac{1}{n}\sum_{i=1}^{n}\|h(w^{(0)};i) - h_i^*\|^2 \cdot \varepsilon$$
$$+ \frac{e^\beta L_\phi}{2\alpha(1-4\alpha)}(4\varepsilon + 3)\left[D^2 H^2 + c(2 + (V + \varepsilon^2 + 2)GD^2)^2 + 2 + V\right] \cdot \varepsilon.$$

$\square$

PROOF OF THEOREM 2

*Proof.* From (42) we have $F_* - \frac{1}{n}\sum_{i=1}^{n}\phi_i(h_i^*) \geq 0$. This leads to

$$\frac{1}{T}\sum_{t=1}^{T}[F(w^{(t)}) - F_*] = \frac{1}{T}\sum_{t=1}^{T}\left(\frac{1}{n}\sum_{i=1}^{n}[f(w^{(t)};i) - \phi_i(h_i^*)] - \left[F_* - \frac{1}{n}\sum_{i=1}^{n}\phi_i(h_i^*)\right]\right)$$
$$\overset{(42)}{\leq} \frac{1}{T}\sum_{t=1}^{T}\frac{1}{n}\sum_{i=1}^{n}[f(w^{(t)};i) - \phi_i(h_i^*)]$$
$$\overset{(45)}{\leq} \frac{e^\beta L_\phi(1+\varepsilon)}{2(1-4\alpha)\alpha\beta} \cdot \frac{1}{n}\sum_{i=1}^{n}\|h(w^{(0)};i) - h_i^*\|^2 \cdot \varepsilon$$
$$+ \frac{e^\beta L_\phi(4\varepsilon + 3)}{2\alpha(1-4\alpha)}\left[D^2 H^2 + c(2 + (V + \varepsilon^2 + 2)GD^2)^2 + 2 + V\right] \cdot \varepsilon. \tag{49}$$

$\square$

PROOF OF COROLLARY 2

*Proof.* For each iteration $0 \leq t < T$, we need to find $v^{(t)}$ satisfying the following criteria:

$$\|v^{(t)} - v^{(t)}_{*\text{ reg}}\|^2 \leq \varepsilon^2,$$

for some $\varepsilon > 0$. Using Gradient Descent we need $\mathcal{O}(n\frac{L}{\mu}\log(\frac{1}{\varepsilon^2})) = \mathcal{O}(2n\frac{L}{\mu}\log(\frac{1}{\varepsilon}))$ number of gradient evaluations (Nesterov, 2004), where $L$ and $\mu = \varepsilon^2$ are the smooth and strongly convex constants, respectively, of $\Psi$. Let

$$\psi_i(v) = \frac{1}{2}\|\eta^{(t)}H_i^{(t)}v - \alpha_i^{(t)}\nabla_z\phi_i(h(w^{(t)};i))\|^2, \; i \in [n]. \tag{50}$$

Then, for any $v \in \mathbb{R}^c$

$$\nabla_v\psi_i(v) = \eta^{(t)}H_i^{(t)\top}[\eta^{(t)}H_i^{(t)}v - \alpha_i^{(t)}\nabla_z\phi_i(h(w^{(t)};i))], \; i \in [n]. \tag{51}$$

Consider $\eta^{(t)} = D\sqrt{\varepsilon}$ for some $D > 0$ and $\varepsilon > 0$, we have for $i \in [n]$ and $0 \leq t < T$

$$\|\nabla_v^2 \psi_i(v)\| = (\eta^{(t)})^2 \|H_i^{(t)\top} H_i^{(t)}\| \leq (\eta^{(t)})^2 \|H_i^{(t)}\| \cdot \|H_i^{(t)}\| \overset{(20)}{\leq} D^2 H^2.$$

Hence, $\|\nabla_v^2 \Phi(v)\| \leq \|\nabla_v^2 \psi_i(v)\| + \varepsilon^2$ for any $v \in \mathbb{R}^c$ which implies that $L = D^2 H^2 + \varepsilon^2$ (Nesterov (2004)) and $\frac{L}{\mu} = \frac{D^2 H^2 + \varepsilon^2}{\varepsilon^2}$. Therefore, the complexity to find $v^{(t)}$ for each iteration $t$ is $\mathcal{O}(2n \frac{D^2 H^2 + \varepsilon^2}{\varepsilon^2} \log(\frac{1}{\varepsilon}))$.

Let us choose $0 < \varepsilon \leq 1$. From (49), we have

$$\frac{1}{T} \sum_{t=0}^{T-1} [F(w^{(t)}) - F_*] \leq \frac{e^\beta L_\phi}{(1 - 4\alpha)\alpha\beta} \cdot \frac{1}{n} \sum_{i=1}^{n} \|h(w^{(0)}; i) - h_i^*\|^2 \cdot \varepsilon$$

$$+ \frac{7e^\beta L_\phi}{2\alpha(1 - 4\alpha)} \left[ D^2 H^2 + c(2 + (V + 3)GD^2)^2 + 2 + V \right] \cdot \varepsilon = N\varepsilon,$$

where

$$N = \frac{e^\beta L_\phi}{(1-4\alpha)\alpha\beta} \frac{1}{n} \sum_{i=1}^{n} \|h(w^{(0)}; i) - h_i^*\|^2 + \frac{7e^\beta L_\phi}{2\alpha(1-4\alpha)} \left[ D^2 H^2 + c(2 + (V + 3)GD^2)^2 + 2 + V \right].$$

Let $\hat{\varepsilon} = N\varepsilon$ with $0 < \hat{\varepsilon} \leq N$. Then, we need $T = \frac{N\beta}{\hat{\varepsilon}}$ for some $\beta > 0$ to guarantee $\min_{0 \leq t \leq T-1} [F(w^{(t)}) - F_*] \leq \frac{1}{T} \sum_{t=0}^{T-1} [F(w^{(t)}) - F_*] \leq \hat{\varepsilon}$. Hence, the total complexity is $\mathcal{O}\left( n \frac{N^3 \beta}{\hat{\varepsilon}^3} (D^2 H^2 + (\hat{\varepsilon}^2/N)) \log(\frac{N}{\hat{\varepsilon}}) \right)$. $\qquad\square$

