# OpenReview forum: "New Perspective on the Global Convergence of Finite-Sum Optimization"
_ICLR.cc/2022/Conference — ICLR 2022 Submitted_

### Official Review · Reviewer_HAWj · 2021-10-22

**Correctness:** 3
**Technical Novelty And Significance:** 3
**Empirical Novelty And Significance:** Not applicable
**Recommendation:** 6
**Confidence:** 4

**Main Review:**

*Strengths*: Overall, the paper is well-written, the problem
is well-motivated, and the background material is sufficient.

*Weaknesses*: Unfortunately, I will have to recommend rejecting
this paper due to a critical mathematical error, namely, **the
constant $V$ used in the proof of Theorems 1-2 does not have the
proper dependence on the tolerance $\varepsilon$**. To elaborate,
notice that the stepsize $\eta^{(t)}$ in Theorems 1-2 is $\Theta(\sqrt{\varepsilon})$
and that Assumption 3 (essentially) states that $V$ is a scalar where
$$
\|v^{(t)}\|\leq V,\quad\|\eta^{(t)}Q^{(t)}v^{(t)}-b^{(t)}\|={\cal O}(\varepsilon),\quad t\geq1,
$$
for matrices $Q^{(t)}$ and vectors $\{v^{(t)}\}$
and $\{b^{(t)}\}$. Even under the generous assumption that
$\{Q^{(t)}\}$ and $\{b^{(t)}\}$ are bounded, the
assumption that $\eta^{(t)}=\Theta(\sqrt{\varepsilon})$ must imply
that, at the very least, $V=\Omega(1/\sqrt{\varepsilon})$. It then
follows the right-hand-side of the bound in equation (18), and similarly in Theorem 2, is no longer $\Theta(\varepsilon)$
but rather $\Omega(1)$ due to the $V^{2}\varepsilon$ term. As a
consequence, **the convergence analysis in the paper is no longer
valid as it can no longer be shown that $\min_{s\leq T}[F(w^{(s)})-F^{*}]=\Theta(\varepsilon)$**

Besides the above major issue, I have a few minor issues with some
of the other material in the paper:

- [p. 4-5] Lemmas 1-2 are classical results and should be replaced
by citations to avoid giving the impression that these are new contributions.
- [p. 5, 6, 14] The use of  "apparently" in several important
statements in this paper implies that these statements have some degree
of ambiguity to them. This kind of wording should be removed if the
authors are making a formal argument, e.g. Appendix C.2, or the authors
should be explicit about their lack of knowledge.
- [p. 6] What is the definition of "smooth" in Remark 1? If
it means continuously differentiable, then clearly $\|J_{w}(\nabla_{w}h_{j}(w^{(t)};I))\|$
can be unbounded even if $w^{(t)}$ is bounded, e.g., consider the
univariate function $h_{j}(w;i)=w^{3/2}$ on $[0,1]$.
- *Minor typos*:
1. [p. 3] ... this paper **focuses** on deep neural networks ...
2. [p. 9] ... for non-convex **problems** (Ghadimi & Lan, 2013) ...
3. [p. 14] ... this result can be **generalized** for larger ...

EDIT: The authors have uploaded a revised version of their manuscript that has addressed my main concerns. The only issues that prevent me from assigning a higher score to the paper are: (i) a lack of numerical experiments and (ii) the use of non-standard assumptions (in particular Assumption 3) that are difficult to verify in practice.

**Summary Of The Paper:**

This paper presents a new optimization method for finding global minima
of nonconvex finite sum problems. In particular, the summands are
functions of the form $\phi_{i}\circ h$ where $\phi_{i}$ is convex
and Lipschitz smooth, while $h$ is nonconvex. Each iteration of the
method consists of solving an auxiliary regularized least squares
(RLS) problem, followed by a gradient step. Additional analysis is
given for when the RLS problem is solved inexactly. Finally, a claimed
$\tilde{O}(\varepsilon^{-3})$ complexity is established under strong
boundedness assumptions on various solution sets.

**Summary Of The Review:**

Despite being a well-written and well-motivated paper, I have to recommend
rejection due to a critical mathematical error. However, if this error
can be resolved in a revision, then I would be open to increasing
my final review score.

EDIT: The authors have uploaded a revised version of their manuscript that has addressed my main concerns. I now
recommend a score of 6 for the paper.

---

> ### Author Response · Authors · 2021-11-12
> **Response to Reviewer HAWj**
>
> We would like to thank you for reviewing our paper and for the thoughtful comments and suggestions. We will revise our paper to make sure there is no confusion and all the typos are fixed.
>
> 1. "**the constant $V$ used in the proof of Theorems 1-2 does not have the proper dependence on the tolerance $\varepsilon$**" and "**must imply that, at the very least, $V = \Omega(1/\sqrt{\varepsilon})$**" and "**the convergence analysis in the paper is no longer valid**"
>
> Thank you for bringing this point and please let us explain it as follows.
>
> First let us assure you that there is no mathematical error in our paper, and we believe that there is a misunderstanding regarding our assumption. Our Assumption 3 essentially assumes that the search direction is bounded by a constant number $V$, and that this number is not dependent on the tolerance $\varepsilon$. Since our main theorems are based on this assumption, our theoretical results remain correct and technically valid.
>
> In addition, we politely disagree with the implication in your review which concludes $V = \Omega(1/ \sqrt{\varepsilon})$, because of the following subtle reason: We cannot conclude $V = \Omega(1/ \sqrt{\varepsilon})$ from the assumption that $Q^{(t)}$ and $b^{(t)}$ are bounded with $\eta^{(t)} = D \sqrt{\varepsilon}$. Notice that if we converge to a global minimum (a stationary point), then the partial derivatives that define matrices  $Q^{(t)}$ and $b^{(t)}$ tend to go to zero. This means that it is not clear how the ratio between the norms of $Q^{(t)}$ and $b^{(t)}$ behave. Therefore, we cannot conclude the behavior of the constant $V$. We also note that there might be more than one solution for the relaxed problem in this assumption, however, we only need that there exists a single solution which is bounded by some $V$. In our paper, Assumption 3 is sufficient to prove the global convergence for non-convex problems in our theoretical framework. Our mathematical treatment has been checked numerous times and every single step in our proof follows a flow without fault.
>
> On the other hand, you might think that this assumption is not standard. However, you may agree that the solution $v_{*}^{(t)}$ plays a similar role as a gradient in the search direction of the (stochastic) gradient descent method (recall that we have the update $w^{(t+1)} = w^{(t)} - \eta^{(t)} v^{(t)}$). It is standard to assume a bounded gradient in the machine learning literature for non-convex problems, and indeed our Assumption 3 is motivated by these facts. For example, for SGD algorithm in Theorem 1 in [Reddi et al. "Stochastic Variance Reduction for Nonconvex Optimization", ICML 2016], the authors also assume $v^{(t)}$ to be bounded by some fixed constant and choose $\eta^{(t)} = O(\varepsilon)$ to guarantee the convergence to a stationary point such that $ \frac{1}{T} \sum_{t=1}^{T} \mathbb{E} [ || \nabla F ( w_t ) ||^2 ] \leq O(\varepsilon)$. We know that the current version of our assumption may be non-standard, however we believe it is acceptable as a part of a new formulation and a new framework for global convergence. It will be very difficult to ask that everything must be standard/traditional while creating new concepts and implementing new ideas. We sincerely believe that you and the research community are open to changes and innovations, and will understand this key point.
>
> 2. “**few minor issues with some of the other material in the paper**”
>
> We thank you for the helpful comments and suggestions. We are also sorry for any confusion in the original version and will revise our drafts carefully to fix these problems. We plan to upload the revised version in the next few days. We will put Lemma 1 and 2 in the appendix and rewrite our arguments in Remark 1 and the appendix sections accordingly without using the word ‘apparently’. We agree with you that some statements can be confused and we clearly do not want to create any more ambiguity.
>
> On the other hand, we are delighted that you find our problem well-motivated, and our contribution significant. When your main concern is resolved (that our paper remains technically correct and valid), we sincerely ask that you reconsider and increase your final review score as promised. We hope that you recognize the significance of the main contributions of our paper as written in the general response.
>
> Thank you very much!

---

> > ### Comment · Reviewer_HAWj · 2021-11-12
> > **Comments about your rebuttal**
> >
> > Thank you for your response.
> >
> > I have carefully read through your argument, but remain unconvinced due to the following reasons:
> >
> > 1. **Bounds on $V$**
> >
> > Consider the following scenario, which immediately rebuts your argument. At $t=0$, suppose that $H_i^{(0)}$ has linearly independent columns, which is fairly reasonable for large networks. Clearly, for every $\varepsilon > 0$, one has that $v_*^{(1)}$ is unique. Moreover, if $||v_*^{(1)}|| = V$ for $\eta = 1$, then for $\eta = D\sqrt{\varepsilon}$ we must have that $||v^{(1)}|| = 1/(D\sqrt{\varepsilon}) = \Omega(1 / \sqrt{\varepsilon})$ by uniqueness. Hence, even in the *first* iteration, we can make $V=\Omega(1 / \sqrt{\varepsilon})$.
> >
> > Furthermore, I am unclear about your argument regarding convergence to a global minimum. The entire point of this assumption on $V$ is to establish global convergence of your proposed method. You cannot assume your conclusion in order to prove the validity of your assumptions that are used to prove your conclusion, i.e., you are using circular reasoning!
> >
> > 2. **[Reddi et. al.]**
> >
> > I believe you are making an improper comparison of the gradients mentioned in [Reddi et. al., 2016] and the size of $v_*^{(t)}$ in your paper. In [Theorem 1, Reddi et. al., 2016], the bounded gradient assumption is regarding the gradient of a differentiable function $f$, which is a fairly mild assumption (it can be satisfied when $f$ is Lipschitz itself), and the computation of $\nabla f$ itself **does not depend on the stepsize $\eta$ nor the tolerance**. In contrast, as reinforced by my argument above, your vector $v_*^{(t)}$ **directly depends on $\varepsilon$ as it is the solution of a least-squares problem *parameterized* by $\varepsilon$**.
> >
> > Again, I appreciate that the authors are pursuing a meaningful and difficult problem using a novel approach, but I cannot recommend accepting any paper to ICLR until I am convinced that the mathematics is correct. We still have 11 days before the end of the rebuttal period, so let us continue to discuss if you believe that I am not understanding your argument.

---

> > > ### Author Response · Authors · 2021-11-12
> > > **New response**
> > >
> > > Thank you for the responsive comments and discussion! We certainly want to clear all your concerns and appreciate that you are helping to improve our paper.
> > >
> > > Firstly, let us clarify that your concerns remain only in Assumption 3 where $V$ is a constant. And as long as we assume this condition holds in our theorems, then our theoretical results remain correct and we have no mathematical error. Could you please help us clarify that?
> > >
> > > Secondly, please let us remind you that we are considering the over-parameterized setting, where the size of the vector $v$ dominates the size of data times the output dimension ($d >> n c$). For this reason, even if we are considering a linear equation, we cannot conclude the linearly independent columns of $H$ and that the solution (in the larger dimension $d$) is unique. In addition, the problem in Assumption 3 is relaxed (i.e. no need to match the exact equation), and for that reason, we can have a larger set of solutions for this problem. Hence, we cannot follow your scenario where there is a unique solution. Therefore, we cannot conclude $V = \Omega(1/ \sqrt{\varepsilon})$.
> > >
> > > [Reddi et. al., 2016] is just an example to motivate our Assumption 3. Moreover, we certainly do not want to introduce circular reasoning. Our example simply shows that we cannot conclude any implications on $V$, since currently we have no clear knowledge about the connections between the two matrices $Q^{(t)}$ and $b^{(t)}$. For these reasons, we believe our assumption (that $V$ is a constant) is still acceptable.
> > >
> > > We really appreciate the time you spend and your willingness to discuss with us. Please let us know if you need any further clarification.

---

> > > > ### Comment · Reviewer_HAWj · 2021-11-12
> > > > **More discussion regarding V**
> > > >
> > > > Indeed, I believe that the theoretical analysis looks fine if we can resolve this single but most crucial issue.
> > > >
> > > > The unique solution setting was just a simple example to give some intuition for my argument, but you are right to point out that this is not the setting in your paper. However, the extension of my argument to the case where the solution set is an affine manifold does not break down as you might think. To elaborate, suppose that $w_*^{(0)} \neq 0$ is a *particular* solution of the least-squares problem at $t=0$ and $\eta^{(0)}=1$ and that the space of solutions is $w_*^{(0)} + U$ for some subspace $U$. Now, suppose $V_{\min{}} := \inf_{v\in w_*^{(0)} + U} ||v||>0$ -- which is reasonable if we want a nonzero direction of descent -- and observe that, specifically at iteration $t=0$, we already can establish $V \geq V_{\min{}}>0$ for $\eta^{(0)}=1$. It now follows that if $\eta^{(0)}=D\sqrt{\varepsilon}$, then $V \geq V_{\min{}} D\sqrt{\varepsilon} = \Omega(1/\sqrt{\varepsilon})$ as claimed.
> > > >
> > > > **Note that we do NOT have to establish a relationship between $Q^{(0)}$ and $b^{(0)}$ in the above argument for it to hold (since these quantities are merely constants)! To reiterate, the argument at $t=0$ does not require any properties of how $Q^{(t)}$ and $b^{(t)}$ evolve over $t$.**

---

> > > > > ### Author Response · Authors · 2021-11-12
> > > > > **New response 2**
> > > > >
> > > > > Thank you for your agreement on our analysis. We also agree with you that we should focus our discussion on a single initial point as you pointed out correctly. However, we believe our assumption is still acceptable, and we are happy to explain.
> > > > >
> > > > > Firstly, we want to note that the solution set in our assumption is not an affine manifold - because the problem is relaxed in terms of $\varepsilon$ as follows (see the right-hand side of the equation):
> > > > >
> > > > > $\frac{1}{2} \frac{1}{n} \sum_{i=1}^{n} || H_i^{(t)} \eta^{(t)}\hat{v}_*^{(t)} - \alpha_i^{(t)} \nabla_z \phi_i (h(w^{(t)};i))  ||^2 \leq \varepsilon^2$
> > > > >
> > > > > Because of this relaxation, our solution set depends on $\varepsilon$, and we cannot simply conclude that $V_{min}$ in your argument is not dependent on $\varepsilon$ and is bounded away from zero. In our assumption, we essentially assume that this solution set (though may depend on $\varepsilon$) will behave nicely enough for us to get a convergence.
> > > > >
> > > > > We thank you for the insightful comments and suggestions that help improve our paper. Your question remind us of another useful discussion, that we are happy to talk here:
> > > > > In the over-parameterized setting of neural networks, in order to prove the convergence, some previous literature actually assumes that the dimension of the vector $v$ can be dependent on the tolerance $\varepsilon$ (see e.g. our reference below). We all agree that when the network is over-parameterized, it has more ability to interpolate all the data and therefore, further relax our problem in Assumption 3. If we allow the network size to vary, then our input matrices may be flexible - at the moment we cannot conclude any particular behavior of the solution set depending on $\varepsilon$.
> > > > >
> > > > > This discussion is not written in our original version, hence we thank you for bringing this up, and we are happy to include these details in our revised draft. Indeed there are more interesting follow up questions regarding the over-parameterized setting, and the overall extent of this paper. That is also the reason why we included over-parameterized settings as related work in our initial draft.
> > > > >
> > > > > For these reasons, we still think our paper has opened up new aspects for the convergence of neural networks as well as non-convex optimization, and it would be unfair if it is denied by a math error - that we carefully investigated to make sure that it did not happen.
> > > > >
> > > > > Thank you for bringing up the insightful discussion so that we can improve the scope of our paper.
> > > > >
> > > > > References:
> > > > >
> > > > > Allen-Zhu et. al., "A Convergence Theory for Deep Learning via Over-Parameterization", ICML 2019
> > > > > (see Theorem 5, non-convex case, the number of neurons $m$ depends on $\varepsilon$.)

---

> > > > > > ### Comment · Reviewer_HAWj · 2021-11-12
> > > > > > **About the inexact case**
> > > > > >
> > > > > > Thank you for the correction regarding the solution set. However, even if we relax the least-squares problem, I do not believe anything will substantially change when going to the inexact case. One can gain an intuition for this by just examining how the solution set behaves for very small values of $\varepsilon$, but for the sake of being precise, I give the formal argument below. Note that indices have been removed to simplify the notation.
> > > > > >
> > > > > > Consider the problem of finding a vector $u$ satisfying $||Qu-b||^{2}\leq\varepsilon^{2}$.
> > > > > > Clearly, this is equivalent to finding a vector $u$ that solves some
> > > > > > (exact) least-squares problem $\min_{v}||Qu-b'||^{2}$ for some $b'\in B(\varepsilon):=${$b'':||b''-b||\leq\varepsilon$}$\cap$
> > > > > > Im($Q$) (to see this, observe that every $u$ in the solution set
> > > > > > of the original problem lies in the intersection of Im($Q$) and a
> > > > > > ball of radius $\varepsilon$ around $b$). Now as long as $b\neq0$,
> > > > > > there will always be $\bar{\varepsilon}>0$ small enough so that $0\notin B(\varepsilon)$
> > > > > > for every $0<\varepsilon\leq\bar{\varepsilon}$, and hence, every
> > > > > > solution of $\min_{v}||Qu-b'||^{2}$ for $b'\in B(\varepsilon)$ and
> > > > > > $\varepsilon\leq\bar{\varepsilon}$ will be bounded away from zero
> > > > > > by a positive scalar (that does not depend on $\varepsilon$). Since finding $v$ satisfying $||D\sqrt{\varepsilon}Qv-b||^{2}\leq\varepsilon^{2}$
> > > > > > is equivalent to finding a vector $u$ that solves the previous problem and scaling $u$ by $1/(D\sqrt{\varepsilon})$,
> > > > > > we once again conclude that $||v||=\Omega(1/\sqrt{\varepsilon})$.
> > > > > >
> > > > > > Regarding your reference, I am not sure if it is related to our discussion. Here, we are fixing the dimension of the network while
> > > > > > computing a descent direction that relates to $\varepsilon$, whereas in [Zhu et al.] they consider varying the dimension according to $\varepsilon$ but computing the descent direction as the gradient of a (fixed) differentiable function.
> > > > > >
> > > > > > I'm afraid I have to disagree regarding the aspect of mathematical errors. If the paper's central claim is both novel and a substantial improvement on the current state-of-the-art, it deserves closer scrutiny. As your paper proposes a first-order algorithm that claims to converge to a global (!) minimum of a non-convex neural network problem (which is notoriously difficult to solve to optimality), it deserves a thorough examination of the mathematics.

---

> > > > > > > ### Author Response · Authors · 2021-11-13
> > > > > > > **New response 3**
> > > > > > >
> > > > > > > Thank you for your careful investigation and thorough comments. We appreciate the time you spend discussing our paper. Please be assured that we also take effort to improve our paper with your suggestion.
> > > > > > >
> > > > > > > We would like to say that we are considering the over-parameterized setting where the dimension $d$ needs to be sufficiently large to interpolate all the data. This observation is also discussed in our paper and for that reason, we do not fix the dimension $d$ to be a constant. Your argument relies on the assumption that the initial matrices must be constant and independent of the tolerance $\varepsilon$, however, it may not always be the case. Note that the initial matrices depend on the initial weight of size $d$ and we can choose this weight depending on $\varepsilon$. When we accept the fact that dimension $d$ could vary and depend on $\varepsilon$ (i.e. we are able to bring the problem to higher dimensions) then our initial matrices could also depend on $\varepsilon$. We will add this discussion to the revised version.
> > > > > > >
> > > > > > > Using the same argument and intuition that you had provided us, if the norm of matrix $H_i^t$ (or matrix $Q$ in your version, of size $c \times d$) is allowed to grow with size $\frac{1}{\sqrt{\varepsilon}}$, where the norm of targeted matrix (or matrix b in your version, of size $c$) remains with constant order, then the learning rate $\eta = D \sqrt{\varepsilon}$ should not be a problem. We think you may feel more comfortable up to this point.
> > > > > > >
> > > > > > > For this reason, we can confirm with you that the analysis of Theorem 1 and our framework for Algorithm 1 holds without any issue.
> > > > > > >
> > > > > > > The problem remains, as you may have noticed, in Assumption 4 and Theorem 2 where we assume that the norm of the matrix $H_i^t$ is bounded, independent of $\varepsilon$. Fortunately we have a simple way to fix this problem - we can relax the assumption with $|| H_i^t|| \leq \frac{H}{\sqrt{\varepsilon}}$ for some $H > 0$ by slightly changing the analysis of Theorem 2 and still get the global convergence. Please allow us a few days to revise the paper and upload a new version. We are excited to improve the scope of our paper, and to convince you of the validation of our assumption.
> > > > > > >
> > > > > > > Our reference is related because it assumes the condition of dimension depending on $\varepsilon$, and it is accepted at ICML, which means that this condition is acceptable in order to get an $\varepsilon$-accurate solution. In the meantime, we understand that it is impossible to find a global minima of a general non-convex problem, without any additional condition. We do not claim that we have any superior magic in our paper, we just claim to provide a new perspective to achieve this goal, and we cannot do this without the support from over-parameterized settings and other conditions in this paper. If you think we overstated somewhere in our draft, please feel free to remind us and we will fix that problem.
> > > > > > >
> > > > > > > Since it is impossible to find the global solution without some additional condition, we sincerely hope that you would accept the over-parameterized condition on the dimension $d$ and the varies of $\varepsilon$ to support our work.
> > > > > > >
> > > > > > > We see that you actually appreciate the motivation behind our paper (to be honest, your appreciation is the main reason why we are having this respectful discussion). Hence we believe you will re-evaluate the revised version and acknowledge the merits of this paper when your main concern is resolved with the acceptance of our assumption. We believe that the new perspective and framework in this submission deserve to get some credit.

---

> > > > > > > > ### Comment · Reviewer_HAWj · 2021-11-13
> > > > > > > > **About your suggestion**
> > > > > > > >
> > > > > > > > Thank you for the additional clarifications. I am excited to see how revision deals with the setting where $H_i^t$ and the size of $v$ may depend non-trivially on $\varepsilon$ (and will definitely raise my score if the analysis looks solid!). I am also fine sticking with the over-parameterized setting for this paper.
> > > > > > > >
> > > > > > > > Regarding your suggested bound $||H_{i}^t|| \leq H / \sqrt{\varepsilon}$, though: as $\varepsilon \to 0$, wouldn't any bound on $||H_i^t||$ trivially hold for sufficiently small enough $\varepsilon$ (and hence, the matrix $||H_i^t||$ is still bounded independent of $\varepsilon$ for sufficiently small enough $\varepsilon$)?

---

> > > > > > > > > ### Author Response · Authors · 2021-11-13
> > > > > > > > > **New response 4**
> > > > > > > > >
> > > > > > > > > Thank you so much for the timely response! We really appreciate your contribution to improve this paper.
> > > > > > > > >
> > > > > > > > > First, please let us clarify that we do not plan to change Assumption 3 or proving it is true for $\varepsilon$. What we are planning to do is implementing the changes as discussed, and adding the discussion that allows the dimension $d$ and the norm of $H_i^t$ to be dependent on $\varepsilon$ (more specifically, $||H_i^t||$ are allowed to grow large with size $\frac{1}{\sqrt{\varepsilon}}$). This change convinces us that Assumption 3 is much more reasonable since the learning rate $\eta = D \sqrt{\varepsilon}$ is no longer a problem.
> > > > > > > > >
> > > > > > > > > Second, regarding your question: we honestly do not really understand your question, though still have the following discussion:
> > > > > > > > >
> > > > > > > > > The matrix  $||H_i^t||$ is still bounded, of course, and this bound may depend on $\varepsilon$. If we are considering the asymptotic behaviour, then this condition is trivial. But if we fix a value of $\varepsilon$ and choose a network that is large enough based on this $\varepsilon$, then the norm of $H_i^t$ is bounded based on $\varepsilon$ and that is our assumption.
> > > > > > > > >
> > > > > > > > > Thank you for giving us this opportunity! We plan to upload the revision early next week.

---

> > > > > > > > > > ### Comment · Reviewer_HAWj · 2021-11-13
> > > > > > > > > > **Minor clarification**
> > > > > > > > > >
> > > > > > > > > > Okay. Be careful with this line of reasoning then.
> > > > > > > > > >
> > > > > > > > > > Just because you allow $||H_i^t||$ to be on the order of $O(1/\sqrt{\varepsilon})$ does not mean it will ever be on that order of magnitude. Specifically, you should be careful with the case of $t=0$ (like in my comments) where $H_i^0$, as it is in the paper, is merely a constant matrix and you can send $\varepsilon\to 0$ without any consequence to $H_i^0$. Ideally, you should think of what assumptions are needed in $H_i^0$, e.g., dependencies on $\varepsilon$ or dimension sizes, and the "$b$" vector to ensure convergence of the method.

---

> > > > > > > > > > > ### Author Response · Authors · 2021-11-13
> > > > > > > > > > > **New response 5**
> > > > > > > > > > >
> > > > > > > > > > > Thank you for your useful suggestions. We will pay attention to these details in the revised version.

---

> > > > > > > > > > > > ### Author Response · Authors · 2021-11-16
> > > > > > > > > > > > **New Revision**
> > > > > > > > > > > >
> > > > > > > > > > > > Dear Reviewer HAWj,
> > > > > > > > > > > >
> > > > > > > > > > > > We have uploaded our revision and implemented the changes as discussed.
> > > > > > > > > > > >
> > > > > > > > > > > > Let us summarize the changes in Theorem 2: Since Assumption 4 is relaxed  (from $||H_t || \leq H$ to $|| H_t || \leq \frac{H}{\sqrt{\varepsilon}}$), now we need a better solution for Algorithm 2 in terms of $\epsilon$ (from $|| v^{(t)} - v_{*\ \text{reg}}^{(t)} ||^2 \leq \varepsilon$ to $|| v^{(t)} - v_{*\ \text{reg}}^{(t)} ||^2 \leq \varepsilon^2$). This fact is shown in equation (19). Hence the total complexity is slightly weaker, from $\tilde{O}(1/\varepsilon^2)$ to $\tilde{O}(1/\varepsilon^3)$. We hope you don’t mind because we still have global convergence.
> > > > > > > > > > > >
> > > > > > > > > > > > Other than the changes made in the general response, more importantly, we reply to you with an example of network initialization. As discussed with you, in order to accommodate the choice of learning rate $\eta^{(t)} = D\sqrt{\varepsilon}$ in our theorems, we describe an example that satisfies $||Q(w^{(0)})|| = \Theta\left(\frac{1}{\sqrt{\varepsilon}}\right)$ where the gradient norm $|| b ||$ is of constant order with respect to $\varepsilon$ (at most).
> > > > > > > > > > > >
> > > > > > > > > > > > We write this example in Section C.3 of Appendix. We would love to hear your feedback and discuss further.
> > > > > > > > > > > >
> > > > > > > > > > > > Other related side notes are:
> > > > > > > > > > > > A similar setting is found in (Allen-Zhu et. al., 2019), where the authors initialize the weights using a random Gaussian distribution with a variance depending on the dimension of the problem. In a non-convex setting, they prove the convergence of SGD using the assumption that the number of neurons $m$ depends inversely on the tolerance $\epsilon$. Our choice of initial weight could be seen as a deterministic version, and intuitively, there could be other random initializations that fit our framework. In that case, our results may hold with high probability, which is similar to recent literature.
> > > > > > > > > > > >
> > > > > > > > > > > > Although our example is simple, it provides some helpful insights on network architecture design and initialization. We want to note that this example can be extended for networks with bias layers and for some certain smooth activation functions. Therefore network design is an interesting potential research question, and that is why we included it in the original draft. We thank you for the motivation and for pushing the discussion forward in this direction.
> > > > > > > > > > > >
> > > > > > > > > > > > We look forward to discussing with you further. We hope this revision resolves your main concern. We believe that this direction is potential and our framework has many new aspects that should be explored in the future.

---

> > > > > > > > > > > > > ### Author Response · Authors · 2021-11-18
> > > > > > > > > > > > > **Revision Discussion**
> > > > > > > > > > > > >
> > > > > > > > > > > > > Dear Reviewer HAWj,
> > > > > > > > > > > > >
> > > > > > > > > > > > > As you may have noticed, phase 1 discussion period ends in three days. Thank you so much for your engagement in this discussion so far. We are looking forward to your response and we sincerely hope to hear back from you soon. Your feedback is important to us, and we want to make sure all your concerns are resolved before November 21. Please allow us a chance to discuss with you and reach an agreement. If you have any suggestion to improve the presentation of our paper, please feel free to let us know before the deadline.
> > > > > > > > > > > > >
> > > > > > > > > > > > > Sincerely,
> > > > > > > > > > > > >
> > > > > > > > > > > > > Authors of Paper 1142

---

> > > > > > > > > > > > > > ### Comment · Reviewer_HAWj · 2021-11-18
> > > > > > > > > > > > > > **Looks good**
> > > > > > > > > > > > > >
> > > > > > > > > > > > > > Dear authors,
> > > > > > > > > > > > > >
> > > > > > > > > > > > > > I am happy with the changes that you have implemented and they have definitely resolved my main concern.
> > > > > > > > > > > > > >
> > > > > > > > > > > > > > Consequently, I have increased my original score of 3 (reject) to 6 (marginal accept) based on your revision and new insights. I have also amended my review in the original post to reflect this change.
> > > > > > > > > > > > > >
> > > > > > > > > > > > > > Regards,
> > > > > > > > > > > > > >
> > > > > > > > > > > > > > Reviewer HAWj

---

> > > > > > > > > > > > > > > ### Author Response · Authors · 2021-11-18
> > > > > > > > > > > > > > > **Thank you!**
> > > > > > > > > > > > > > >
> > > > > > > > > > > > > > > Dear Reviewer HAWj,
> > > > > > > > > > > > > > >
> > > > > > > > > > > > > > > Thank you very much for your support! Your appreciation means a lot to us and our discussions give us more insights and motivation to expand this research direction. We believe that this direction is potential and our framework has many new aspects to explore.
> > > > > > > > > > > > > > >
> > > > > > > > > > > > > > > We are happy to see that your main concern has been resolved, and we thank you for the time and effort you have spent to improve this paper.
> > > > > > > > > > > > > > >
> > > > > > > > > > > > > > > Sincerely,
> > > > > > > > > > > > > > >
> > > > > > > > > > > > > > > Authors of Paper 1142

---

### Official Review · Reviewer_ZKUZ · 2021-11-01

**Correctness:** 3
**Technical Novelty And Significance:** 2
**Empirical Novelty And Significance:** Not applicable
**Recommendation:** 6
**Confidence:** 5

**Details Of Ethics Concerns:**

There are no concerns.

**Main Review:**

Strengths: the design of the algorithms based on the particular formulation used; the underlying analysis

Weaknesses: While this is a theory oriented paper, the theoretical portion is not strong enough to compensate lack of an experimental study. I am unconvinced about the novelty of the analyses. The authors should provide more convincing arguments that the entire material is unrelated to derivate-free second order algorithms and for example, BFGS.
Regarding Theorem 1, something is strange with respect to \beta. If beta is large (goes to infinity), it increases the number of iterations, yet the right-hand-side in (18) goes to infinity. It should be the other way around.
To approximately solve (15) or its original version, I wonder why the authors don't consider conjugate gradient descent. It works well for quadratic problems.

**Summary Of The Paper:**

The paper provides a new gradient-based algorithm. The algorithm is based on the observation that a loss function for a single sample can be written as composition of two functions (the logits and the actual loss function). It computes the direction by means of solving a quadratic MSE problem. They provide a convergent analysis of the algorithm (there is a version where the quadratic problem is solved explicitly through a closed form expression and a version where gradient descent is applied to solve the problem approximately). There are no computational experiments.

The main contributions are in the algorithms themselves and the accompanying analyses. It's unclear how novel are the proof techniques since everything resembles second order algorithms. The authors also claim the actual reformulation to be a novel contribution however such a formulation is straightforward and used in many contexts (some of my lecture slides from years ago show the formulation used by authors as a possible formulation for the overall loss function). Despite of this, the design of the algorithm should get credit.

**Summary Of The Review:**

I give authors credit for the algorithmic development and to a certain degree analyses. They are over-selling the work by using 'novel' too often (in particular in connection with the formulation). The role of \beta in Theorem 1 makes me worried about the significance of the result.

In short, this is nice work but not impressive. Addressing some of the weakness would definitely improve the paper.

Other remarks:
Remark 1: w^t stay in a bounded region: This is often not the case in practice. Gradient clipping is often needed to circumvent this. The reLU issue is addressed in an ad-hoc manner.
Assumption 3: On the first read, it felt like such a \hat{v}_aster^t always exists. Then I realized that it is \hat{v}_aster^t(\epsilon). The authors should probably emphasize this dependency.

Minor comments:
The "Loss Functions" subsection should be take out. It's trivial. Definitions 2 and 3 should be removed as it is safe to assume that a reader not knowing these concepts would not encounter this paper (even less being able to understand it).
New Algorithmic Framework
Period missing at the end of (7)
It is weird to write H (matrix) \cdot \eta (scalar) \cdot v (vector). I suggest \eta H v
The assumptions in Theorem 2 don't have to be repeated. One can simply state "under the same assumption as in Theorem 1 plus additional assumptions ..."

---

> ### Author Response · Authors · 2021-11-12
> **Response to Reviewer ZKUZ (1/2)**
>
> We thank you for your constructive comments and the positive score on our paper. We hope our answers below address all your concerns.
>
> 1. "**The authors also claim the actual reformulation to be a novel contribution however such a formulation is straightforward and used in many contexts**" and "**how novel are the proof techniques**"
>
> We notice that our problem formulation is $min_{w} \frac{1}{n} \sum_{i=1}^{n} \phi_i ( h ( w ; i ) )$, where $\phi_i$ and $h (w ; i)$ are dependent on $i$. Moreover, for all $i = 1,\dots,n$, the function $h ( w; i )$ shares the same parameter $w$.
>
> At a first glance, our formulation may seem to be straightforward to build, and similar to previous contexts. However, we have a thorough discussion at the beginning of page 3 in our paper explaining the difference from existing composite problems in e.g. (Zhang & Xiao, 2019; Tran-Dinh et al., 2020). Our problem is fundamentally different from these works since they use a common outer function $\phi$. Our formulation has the dependency on the data $i$ for both inner and outer functions, which makes the analysis become non-trivial. This subtle difference motivates our framework and design of algorithms in our paper.
>
> Without the separate convex functions $\phi_i$, we would not be able to come up with the key insights that lead to Framework 1 in Section 4.
> Moreover, by taking the advantages of the new formulation structure, we are able to obtain the global convergence results for non-convex problems while it is unclear how to derive global convergence analysis in the non-convex cases for the previous formulations.
>
> We have tried our best to check the related literature, but still we are not aware of existing work that considers the exact same formulation. We would really appreciate it if you could provide us the reference so that we can cite them properly.
>
> 2. "**It's unclear how novel are the proof techniques since everything resembles second order algorithms**" and "**The authors should provide more convincing arguments that the entire material is unrelated to derivative-free second order algorithms and for example, BFGS**"
>
> Our proof technique is fundamentally different from the one of the second order algorithms. First, our problem setting is different as mentioned above. With this dependency on the data $i$, the problem becomes non-trivial to solve. Moreover, we do not use any second order information or try to approximate Hessian in our algorithm. It is unclear how to apply second order methods or BFGS into our problem. Second, we are able to show the global convergence for our problem setting while the second order methods may not. For these reasons, we think that our proof techniques are unrelated to the second order type algorithms. We will add this discussion in the revised version.
>
> 3. "**theoretical portion is not strong enough to compensate lack of an experimental study**"
>
> We emphasize that our main contribution is the development of a new theoretical framework. Indeed, you agreed that our work should get credit for the algorithmic development. Thus, our most important contribution is the new perspective and theoretical framework that motivates further study and optimization of the new algorithms for global convergence for non-convex problems. Designing a class of algorithms that can guarantee global convergence without strong assumptions on the objective function is hard. Our framework shows a new way of thinking to achieve this goal. Please also refer to our general response for further details. We believe that our theoretical foundation is strong and significant for publication.
>
> 4. "**If beta is large (goes to infinity), it increases the number of iterations, yet the right-hand-side in (18) goes to infinity**" and "**The role of \beta in Theorem 1 makes me worried about the significance of the result**"
>
> We are sorry for the confusion. Actually we left $\beta$ as a flexible parameter just for the general purpose of choosing $T$. We can simply choose $\beta = 1$ with $T = \frac{1}{\varepsilon}$ and replace the whole analysis by $\beta = 1$. Our result still holds without any issues. We will add an explanation to the revised version.

---

> > ### Author Response · Authors · 2021-11-12
> > **Response to Reviewer ZKUZ (2/2)**
> >
> > 5. "**I wonder why the authors don't consider conjugate gradient descent**"
> >
> > Thank you for your suggestion! Indeed, the conjugate gradient descent is also a nice choice to solve (15). Actually, our main goal is to provide an algorithm for approximately solving (15), and gradient descent is a simple one, in order to show the global convergence of problem (1). In fact our Framework 1 is flexible and general enough to consider different optimization methods, and this is a strength of our paper. We will have more discussions on using other algorithms such as conjugate gradient descent as you suggested.
> >
> > 6."**Minor comments and suggestions**"
> >
> > We thank you for the helpful comments and suggestions. We are also sorry for any confusion in the original version and will revise our drafts carefully to fix these problems. We will rewrite Section 3 as you suggested and revise our writing in Remark 1, Assumption 3 and Theorem 2 accordingly. We plan to upload the revised version in the next few days.
> >
> > We thank the reviewer for constructive comments. We will revise our manuscript based on your comments. We hope that our answers address all your concerns, and sincerely ask that you help us convince other reviewers that our algorithmic design and theoretical contribution should get credit.

---

> > > ### Author Response · Authors · 2021-11-16
> > > **New Revision**
> > >
> > > Dear Reviewer ZKUZ,
> > >
> > > We have uploaded our revision and implemented the changes as discussed with you. We thank you for the useful suggestions that help improve the presentation and expand the scope of our paper.
> > >
> > > We added related discussions and clarification for our formulation in Section 2. We shortened Section 3 and moved other materials to the appendix, for a better presentation of the background materials. We replaced Remark 1 by a discussion about second-order methods, and added explanations for $\beta$ and conjugate gradient descent. Since the statement of two theorems are slightly different, to be mathematically clear, we keep them as the previous version.
> > >
> > > For your convenience, the main changes are highlighted in the revision.
> > >
> > > We invite you to read the revision along with our responses. We hope to address all your concerns and please do not hesitate to contact us if you need any additional clarifications.

---

### Official Review · Reviewer_eoVW · 2021-11-02

**Correctness:** 2
**Technical Novelty And Significance:** 2
**Empirical Novelty And Significance:** 2
**Recommendation:** 6
**Confidence:** 4

**Main Review:**

The submitted work presents some interesting result, such as using the composite structure for machine learning loss functions, and approximations approach to determine the descent direction. However, I have some concerns,
1. The formulation is not new. This is a quite well studied formulation in the optimization community. For instance, the authors can check the work of Lewis and Wright 2016. For their result, non-smooth outer functions can be allowed. So the proposed approach is not novel, and the authors should discuss these existing work.
2. The obtained complexity result is not strong, as it is in the sense of
$$\min_{0 \leq t\leq T-1} [F(w^t) - F_*] \leq \frac{1}{T} \sum_t [ F(w^t)-F_* ]  = O(\epsilon) . $$ Given the good properties of the problem.
3. It would be great if the authors can provide numerical experiments, to demonstrate the performance of the proposed work.

The author addressed my concerns fairly, as a result i raised my score by 1 to 6.

**Summary Of The Paper:**

The submitted paper considers a composite formulation of machine learning loss functions, where both the inner and outer functions are required to be smooth. Two algorithms are proposed for the considered formulation, with corresponding convergence results.

**Summary Of The Review:**

The paper provides certain interesting result, however the novelty of the paper and advantage of the proposed algorithm need further justification.

---

> ### Author Response · Authors · 2021-11-12
> **Response to Reviewer eoVW**
>
> We thank you for your thoughtful review and comments. We look forward to discussing with you and answering your questions regarding the novelties and contributions of our paper.
>
> 1. "**The formulation is not new**" and "**authors can check the work of Lewis and Wright 2016**"
>
> Let us clarify how our formulation is different from Lewis and Wright, 2016: The problem considered in Lewis and Wright, 2016, is $min_{x} h ( c ( x ) )$, while our problem formulation is $min_{w} \frac{1}{n} \sum_{i=1}^{n} \phi_i ( h ( w ; i ) )$, where $\phi_i$ and $h (w ; i)$ are dependent on $i$. Also, each function $h ( w; i )$, $i= 1, \ldots, n$,  shares the same parameter $w$. Our problem is expressed as a finite sum of dependent composition functions, hence it is unclear how to rewrite it in terms of the composition $h(c(x))$ of some universal functions. Therefore the problem in Lewis and Wright, 2016, does not cover our problem, and as a consequence the techniques used by Lewis and Wright cannot be applied to our problem.
>
> In addition, at the beginning of page 3 in our paper we have a discussion about the existing composite problems in (Zhang & Xiao, 2019; Tran-Dinh et al., 2020). We explain how our formulation is fundamentally different from these works since they use a common/universal function $\phi$ independently from $h ( w; i )$. In our formulation $\phi_i$ depends on the data $i$, making our problem non-trivial to solve. In addition, we notice that it is unclear how to derive a global convergence analysis for the previous formulations, while we are able to obtain global convergence results for non-convex problems in our theoretical and algorithmic framework.
>
> The reason why we choose to investigate smooth and convex outer functions is that we focus on the two well-known loss functions corresponding to classification and regression tasks in deep learning as shown in Lemmas 1 and 2.
>
>
> 2. "**The obtained complexity result is not strong**"
>
> We notice that our problems are generally non-convex. The common convergence criteria for finding a **stationary point** for non-convex problems is $ \frac{1}{T} \sum_{t=1}^{T} || \nabla F ( w_t ) ||^2 \leq O(\varepsilon)$, which has been widely used in the existing literature for non-convex optimization problems. Therefore, we believe that the convergence criteria $\frac{1}{T} \sum_{t=1}^{T} [ F (w_t) - F* ] \leq O(\varepsilon)$ for finding a **global** solution for non-convex problems is significant. Given that our algorithms can find a global optima, the derived complexity results are not weak, compared to relevant existing methods in literature.
>
> 3. "**it would be great if the authors can provide numerical experiments, to demonstrate the performance of the proposed work**"
>
> For this question, we refer to our elaborate discussion in our general response.
>
> We thank the reviewer for their constructive comments. We hope our answers help explain our contributions. If you have further concerns please reach out during the discussion period. We hope that our answers address all your concerns and you will consider re-evaluation of our work.

---

> > ### Comment · Reviewer_eoVW · 2021-11-15
> > **Response**
> >
> > Thank you for your answers, and sorry for this late responses.
> >
> > I appreciate the replies which look good to me. Just to clarify the 1st point, which I didn't make myself clear in the review. By not new, I meant to say that the proposed formulation is extending the form of Lewis and Wright to the finite sum case. That is
> > $$\min h(c(x)) \to \min \frac{1}{m} \sum_i h(c(x; i)) . $$
> > And for loss function $\phi_i$ depending on $i$, I could not agree with this statement. Since loss functions are basically quadratic loss, cross-entropy and hinge loss, etc, as mentioned in Section 3. The values of loss function depends on $i$, but its formulation does not. Only if, for example both square loss and cross-entropy loss appear in the model in the same time, then I agree $\phi_i$ depends on $i$. But I don't think this is realistic and is the case considered in the paper.

---

> > > ### Author Response · Authors · 2021-11-15
> > > **New response 1**
> > >
> > > Thank you for your appreciation! We are delighted to hear about your perspective and to discuss with you further.
> > >
> > > We agree with you that our formulation considers the composition related to Lewis and Wright for the finite sum case. We only want to note that these two formulations are very different in terms of the methods we are using to solve them. For example, a class of variance-reduced algorithms (e.g. SAG/SAGA/SVRG/SARAH) is designed specifically for the finite-sum formulation in order to show the benefits over Gradient Descent. Additionally, though the finite sum form $\sum_i f_i(w)$ is a special case of the general one $F(w)$ when $F(w) = \sum_i f_i(w)$; we cannot do the same for the ‘general’ $h (c(w))$ in our formulation unless $h$ is identical function (which sadly is not helpful).
> > >
> > > Regarding your comments with the loss function $\phi_i$:
> > >
> > > Firstly we want to note that our formulation is not restricted in the ML setting. We can consider other applications that fit our model and allow different loss functions. Thank you for pointing this out.
> > >
> > > Secondly, please let us persuade you that the loss function $\phi_i$ is actually natural for our setting. We already know that $h(w;i)$ is the output of model, and the loss function (square loss/ CE loss) is a “distance measure” between our output $h(w;i)$ and the targeted output $y^{(i)}$. (e.g. in square loss $\phi_i (output) = ||output - y^{(i)} ||^2$).
> > >
> > > We want to minimize this “distance”, and though the name of the loss function is the same, we surely do not want to have the same function $\phi$ for every data output $h(w;i)$. We want it to be $\phi_i$ so that it can try to fit our output to a specific target $y^{(i)}$, and this is different for each data where $y^{(i)}$ has different values. And that is the main reason why we have different $\phi_i$ instead of a common function. This fact actually reflects the natural setting of ML where the outputs are designed to fit different targets, which makes the problem nontrivial.
> > >
> > > We think this discussion is informative and very helpful for future readers to thoroughly understand our paper. We plan to add this to the revision with some other changes suggested by the reviewers. We invite you to check the revised version in the next few days. When all your concerns are resolved, we sincerely hope that you could re-evaluate the paper and consider increasing your score.
> > >
> > > Thank you!

---

> > > > ### Comment · Reviewer_eoVW · 2021-11-16
> > > > **Response 2**
> > > >
> > > > Thank you for your replies, they are looking good to me!

---

> > > > > ### Author Response · Authors · 2021-11-16
> > > > > **New Revision**
> > > > >
> > > > > Dear Reviewer eoVW,
> > > > >
> > > > > We have uploaded our revision and implemented the changes as discussed. We thank you for the useful discussion and perspective that help improve the presentation and readability of our paper.
> > > > >
> > > > > We added your reference (Lewis and Wright, 2016) with related discussions and clarification for our formulation in Section 2. We also add a discussion regarding the convergence criteria after Theorem 1 as discussed with you.
> > > > >
> > > > > For your convenience, the main changes are highlighted. We are sorry that we cannot highlight citations.
> > > > >
> > > > > We invite you to read the revision along with our responses. We hope to address all your concerns and please do not hesitate to contact us if you need any additional clarifications.

---

> > > > > ### Author Response · Authors · 2021-11-19
> > > > > **Revision Discussion**
> > > > >
> > > > > Dear Reviewer eoVW,
> > > > >
> > > > > Thank you for your encouragement! We are delighted that you agree with our responses. We have implemented the changes and added the detailed explanations as discussed with you. We sincerely hope that this revision resolves all of your concerns. If you have any suggestion for us to improve the presentation of our paper, please let us know before November 21st so that we have time to answer you.
> > > > >
> > > > > Since our paper presents a new approach, and you actually appreciate our answers, it is really important for us to have your support. Our revision is much better now thanks to the suggestions by the reviewers. We would be grateful if you could re-evaluate our paper and consider increasing your score. We look forward to hearing back from you.
> > > > >
> > > > > Thank you for your participation and the time you spent reviewing this paper!
> > > > >
> > > > > Sincerely,
> > > > >
> > > > > Authors of Paper 1142

---

### Official Review · Reviewer_mfHX · 2021-11-03

**Correctness:** 4
**Technical Novelty And Significance:** 2
**Empirical Novelty And Significance:** Not applicable
**Recommendation:** 6
**Confidence:** 2

**Main Review:**

Strengths: the paper is fairly clear, and the proposed transformation is certainly a novel way to solve a common class of problems in the machine learning community.

Weaknesses: I'm finding it extremely difficult to evaluate the significance of this work.  I respect that there's value in providing a new perspective on an old problem (with correspondingly different asymptotic bounds on its performance), but unless this proposed method is demonstrably useful at solving a concrete problem, I'm not sure if it's significant enough for ICLR.  Specifically, the version of this paper that I would probably feel good about accepting would kick all of sections 3, 4, and 5 to appendices, would succinctly state Algorithms 1 and 2, and then would spend the rest of the paper evaluating these algorithms on real problems.

At the same time, I don't want to discourage this kind of work, so I would honestly be satisfied seeing this proposed algorithm applied to _any_ problem, but as it stands, I don't think I can accept.

edit: raising score to 6 after author's updates---still would like to see empirical analysis.

**Summary Of The Paper:**

The authors formulate a way to transform finite-sum optimization problems in a proxy strongly convex problem, and prove that it converges to a global minimum in a number of gradient steps that scales inverse quadratically with the tolerance.

**Summary Of The Review:**

The authors propose an interesting and novel algorithm for machine learning, but it's unclear how significant it will be.

---

> ### Author Response · Authors · 2021-11-12
> **Response to Reviewer mfHX**
>
> We thank you for your valuable comments and suggestions! We appreciate that you acknowledge our work as a new perspective and ‘a novel way to solve a common class of problems in the machine learning community’. We hope our answers below address all your concerns.
>
> 1. "**extremely difficult to evaluate the significance**" and "**I'm not sure if it's significant enough for ICLR**"
>
> The class of problems we are choosing to investigate is generally non-convex. For the broad class of non-convex problems it is almost impossible to find an efficient class of algorithms that can guarantee global convergence (without strong assumptions on the objective function). Most first-order methods in literature, e.g. SGD and their variants, can only find a **stationary point** (under the convergence criteria of $ \frac{1}{T} \sum_{t=1}^{T} || \nabla F ( w_t ) ||^2 \leq O(\varepsilon)$. Thus providing a new theoretical foundation with an algorithmic framework that offers a global convergence analysis for non-convex problems is very impactful.
>
> Our framework in Section 4 allows us to investigate a new class of algorithms that can find a **global** minimizer. We note that global convergence is very difficult to reach for non-convex problems, and we are able to achieve this thanks to the structure of our new formulation. For that reason, our convergence criteria $\frac{1}{T} \sum_{t=1}^{T} [ F (w_t) - F* ] \leq O(\varepsilon)$ for non-convex problems is also different from previous works.
>
> Finally, we study the global convergence of deep neural network problems, where we consider softmax cross-entropy loss and square loss for classification and regression problems, respectively, in an over-parameterized setting. Finding a global solution of the optimization problem for training deep neural networks fits our theoretical framework and therefore significantly contributes to the ICLR community.
>
>
> 2. "**I would probably feel good about accepting would kick all of sections 3, 4, and 5 to appendices**"
>
> Thank you for your suggestions. We agree with you and other reviewers that Section 3 should be shortened and some details should be moved to the appendix for a better presentation. However, Section 4 presents our key insights and motivations that lead to the development of a new framework and two algorithms as well as their convergence results as shown in Section 5. We note that our approach is non-trivial since it is designed to exploit the structure of the new formulation and the outer convex loss function. Therefore, we believe that Sections 4 and 5 are important and need to be kept in the main content.
>
> 3. "**I would honestly be satisfied seeing this proposed algorithm applied to any problem**"
>
> For this concern, we refer to our elaborate discussion in our general response. We do not expect the two presented algorithms that fit our framework to be efficient and practical -- optimization and detailed experimentation is needed in order to achieve this. Also, our framework is general in that it fits other algorithmic solutions that can lead to practical implementations in future work.
>
> We thank the reviewer for their constructive comments. We hope our answers help explain our contributions. If you have further concerns please reach out during the discussion period. We hope that our answers address all your concerns and you will consider re-evaluation of our work.

---

> > ### Author Response · Authors · 2021-11-16
> > **New Revision**
> >
> > Dear reviewer mfHX,
> >
> > We have uploaded our revision and implemented the changes as discussed with you. We thank you for your suggestions that help improve the presentation of our paper.
> >
> > We added related discussions and clarification for our formulation in Section 2. We shortened Section 3 and moved other materials to the appendix, for a better presentation of the background materials.
> >
> > For your convenience, the main changes are highlighted in the revision. We invite you to read the revision along with our response. We hope to address all your concerns and you will consider re-evaluation of our work.

---

> > > ### Author Response · Authors · 2021-11-19
> > > **Revision Discussion**
> > >
> > > Dear Reviewer mfHX,
> > >
> > > Thank you for your appreciation that our algorithms are interesting and novel. Since November 21st is approaching, we want to get back to you and see if you have any remaining concerns. We have improved our paper using the suggestion from you and other reviewers. Our previous concern with another reviewer has been resolved recently. Similarly, we hope to hear your response before the deadline, and if possible, clear up any remaining unclarity about our paper.
> > >
> > > As you wrote in your review, our main contributions are the new theoretical and algorithmic framework. In fact, based on our theoretical foundation, there is still room for improvement and for better design of practical algorithms in the future. We believe that this research direction is potential and our framework has many new aspects to explore further. Indeed the discussion with the reviewers helped us improve our paper.
> > >
> > > For that reason, your support and appreciation is really important to us. We believe our new version is now better and we sincerely hope you could re-evaluate the revision and support our new framework.
> > >
> > > Thank you for the time and effort you have spent reviewing this paper!
> > >
> > > Sincerely,
> > >
> > > Authors of Paper 1142

---

> > > > ### Comment · Reviewer_mfHX · 2021-11-19
> > > > **Response to response**
> > > >
> > > > In the interest of encouraging research work that is more foundational / where the grain size of "success" is not always package-able into a unit of "SOTA on a task people care about", and in appreciation of the authors significant efforts to resolve issues raised by all of the other reviewers, I'm raising my score to a 6.
> > > >
> > > > That said, I fundamentally disagree with the authors that an empirical investigation of the proposed algorithm is not essential to this paper.  Even for algorithms that are not "ready for prime time", a toy empirical demonstration that the algorithm actually works on a task resolves several problems simultaneously:
> > > > 1) It provides a sanity check that the asymptotics proved by the authors are valid
> > > > 2) It gives tangible substance to how the algorithm works by connecting the alphabet soup of constants appearing in, e.g. Eq. 18, to intuitively accessible facets of a real problem
> > > > 3) It (possibly) clarifies _what remains to be done_ to bring the algorithm from its nascent state to something potentially practically useful.
> > > > 4) It situates the algorithm (however fairly) with respect to competing approaches in the literature
> > > >
> > > > Not considering _any_ of these issues makes it difficult to know if the author's proposed approach will have _any_ future payoff.  Admittedly, this is the nature of some research: that it is difficult to know if it will pay off.  But completely avoiding this kind of analysis only hurts.

---

> > > > > ### Author Response · Authors · 2021-11-19
> > > > > **Thank you!**
> > > > >
> > > > > Dear Reviewer mfHX,
> > > > >
> > > > > Thank you so much for your support and your appreciation of our effort! Indeed we are working hard to improve the scope of our paper, and practical implementation and evaluation of algorithms is the certain thing we will consider in the near future. We appreciate your perspective and we thank you for providing the helpful comments and valuable insights for our upcoming empirical investigation.
> > > > >
> > > > > We totally agree that it is difficult to know if a particular approach will have any payoff - and this is the reason why your acceptance is important to our work. This support from you and other reviewers will be our motivation to pursue and explore this direction further - both theoretically and empirically.
> > > > >
> > > > > Thank you so much,
> > > > >
> > > > > Authors of Paper 1142

---

### Author Response · Authors · 2021-11-12
**General response**

First of all, we would like to thank the AC and all other reviewers for your hard work and for reviewing our paper.

**To all reviewers**: Below we first highlight our contributions in order for you to have a better overview of our paper, after which we clarify current misunderstanding and/or confusion around our work. By addressing your concerns we hope you will be motivated to discuss our paper with us and clear up any remaining unclarity.

The class of problems we are choosing to investigate is generally non-convex, and for this reason, it is almost impossible to find an efficient class of algorithms that can guarantee global convergence (without strong assumptions on the objective function). Our problem formulation is $min_{w} \frac{1}{n} \sum_{i=1}^{n} \phi_i ( h ( w ; i ) )$, where we note that $\phi_i$ and $h (w ; i)$ are dependent on $i$. Moreover, for all $i = 1,\dots,n$, the function $h ( w; i )$ shares the same parameter $w$. At first glance, our formulation may seem straightforward, and similar to previous contexts. However, our paper has a thorough discussion showing the difference from existing composite problems in e.g. (Zhang & Xiao, 2019; Tran-Dinh et al., 2020). Our problem is fundamentally different from these works since they use a common outer function $\phi$. Our formulation has the dependency on the data $i$ for both inner and outer functions, which makes the analysis become non-trivial. This subtle difference allows us to derive new properties of the objective function which leads to our framework and design of algorithms as you have seen in this paper. We notice that it is unclear how to derive global convergence for non-convex problems in previous formulations while in our algorithmic framework we are able to do this.

We emphasize that our main contributions are in the form of a new developed theoretical and algorithmic framework. Indeed, the reviewers agreed that our work provides a new perspective and a novel way to solve a common class of problems in the machine learning community. We stress that not every idea in this ML community is considered practical right at the time it was proposed -- our most important contribution is the new perspective and theoretical framework that motivates further study and optimization of the new algorithms for global convergence.

We believe that experiments are not essential to our paper. In particular, we do not claim that the current version of our two algorithms work efficiently in practice. In fact, based on our theoretical foundation, we believe there is still room for improvement and for better design of practical algorithms in the future. Hence, a translation to a practical implementation with empirical results should be left to follow-up work. Indeed, we hope that, in addition to broadening our theoretical understanding of the global convergence for DNNs, our paper will inspire others to implement and experiment with our framework.

This general response together with our individual responses to each reviewer address the reviewers' concerns. Given the significance of our theoretical and algorithmic framework which provides global convergence for non-convex problems, we sincerely encourage the reviewers to re-evaluate our work and consider increasing their scores.

Please do not hesitate to contact us if there are additional clarifications that we can make to convince the reviewers of the significance of our paper. We appreciate AC’s effort and time to handle the review discussion and we also value all the reviewers’ feedback and comments. We would love to receive more follow-up responses and give clarifications to help the discussion of our work.

Sincerely,

Authors of Paper 1142

---

### Author Response · Authors · 2021-11-16
**Revision**

Dear everyone,

We have uploaded our revision and implemented the changes as discussed with the reviewers. We thank everyone for the useful suggestions that help improve the presentation and expand the scope of our paper.

For your convenience, the main changes are highlighted in the revision. We summarize these changes as follows:

1. We added related discussions and clarification for our formulation in Section 2. We shortened Section 3 and moved other materials to the appendix, for a better presentation of the background materials. We replaced Remark 1 by a discussion about second-order methods and implemented other minor changes and made sure there is no confusion.

2. Assumption 4 is relaxed to have a better dependence on $\varepsilon$ (specifically, from $||H_t || \leq H$ to $|| H_t || \leq \frac{H}{\sqrt{\varepsilon}}$). This change is made thanks to the discussion and suggestion by Reviewer HAWj, to accommodate better with the dependence of $\varepsilon$ in Assumption 3. Our algorithm and analysis are slightly modified and the complexity changes from $\tilde{O}(1/\varepsilon^2)$ to $\tilde{O}(1/\varepsilon^3)$. We believe this is not a problem because we still have global convergence.

We are sorry that some citations and equations cannot be highlighted. Since we cannot modify our paper after November 22, we will remove the highlights right before that date.

We invite you to read the revision along with our responses. We hope to address all your concerns and please do not hesitate to contact us if you need any additional clarifications. We would love to receive further feedback from you.

Sincerely,

Authors of Paper 1142

---

### Author Response · Authors · 2021-11-22
**Thank you!**

Dear everyone,

We thank all the reviewers for their constructive comments and suggestions that help improve our paper.

We are delighted to hear that you all agree to support us, and we are truly grateful for your substantial help during the discussion period. The highlights in our revision have been removed since we cannot modify the paper after November 22th.

Again, we thank the Area Chair and all other reviewers for your hard work and for reviewing our paper.

Sincerely,

Authors of Paper 1142

---

### Public Comment · ~Lam_M._Nguyen1 · 2022-02-02
**Decision**

Dear all,

First of all, we would like to thank the reviewing committee for your useful comments and suggestions to improve our paper.
Although we accept the final decision, we respectfully disagree with the implications in this meta-review. To be more specific:

The interpretation of Assumption 3 is not necessarily correct. From our assumption, we cannot conclude that $H_i * v \approx \nabla_z \phi_i (h(w; i))$ as mentioned in the meta-review. The implication that “$||\nabla_w f(w;i)||$ is as large as $|| \nabla_z \phi_i (h(w; i)) ||^2$” is also vague, and even if it is true, it does not imply that gradient descent will work under our assumption. Note that for general non-convex setting, the convergence criteria uses the squared norm of full gradient $|| \nabla_w F(w) ||^2$ (e.g. see our Remark 2), not the squared norm of individual gradient $|| \nabla_w f(w;i) ||^2$.

For these reasons, it is not clear that (stochastic) gradient descent can achieve convergence to a global solution for nonconvex setting, even with our assumption.

Sincerely,

Authors of Paper 1142

---

### Decision · Program_Chairs · 2022-01-20

**Decision:**

Reject

**Comment:**

This paper proves a global convergence rate of a newly proposed algorithm finite sum problem under some assumptions. While the proposed algorithm provides some interesting ideas to solving the finite sum problem using intermediate proxy solver, the current assumptions are too strong and I'm afraid that this can make the result essentially trivial:

For example, assumption 3 assumes that
$ H_i * v \approx \nabla_z\phi_i(h(w, x_i))$ for every i. This simply implies that
$ \|\nabla_w\phi_i(h(w, x_i)) \|_2 $ is as large as $\|\nabla_z\phi_i(h(w, x_i))\|_2^2 $ as long as the norm of v is small, since
$\nabla_w\phi_i(h(w, x_i)) =  \nabla_z\phi_i(h(w, x_i)) H_i $.



Hence the assumption simply assumes that "If the loss is not small, then the gradient of the objective is not small (using the convexity of $\phi$, so $|\nabla_z\phi_i(h(w, x_i))$ has to be large)" -- This would imply that gradient descent can also work (and arguably having the same convergence rate) under this assumption.  Note that "the smallest movement that can decrease the objective the most" is indeed following gradient descent direction -- So gradient descent would not move the weights more than this algorithm as well.

Therefore, I am not sure that there is a clear benefit to using this algorithm compared to the standard (stochastic) gradient descent. In particular, I would suggest the authors at least show one example where under the current set of assumptions, gradient descent does not work as efficiently compared to the proposed algorithm -- This will make the proposed algorithm much more justified.